# Transport and inhibition mechanism of the human SGLT2–MAP17 glucose transporter

**Masahiro Hiraizumi** [1,4,5] ✉, **Tomoya Akashi** [2,5], **Kouta Murasaki**[1], **Hiroyuki Kishida**[1], **Taichi Kumanomidou**[1], **Nao Torimoto**[1], **Osamu Nureki** [3] ✉ & **Ikuko Miyaguchi** [1] ✉

Sodium–glucose cotransporter 2 (SGLT2) is imporant in glucose reabsorption. SGLT2 inhibitors suppress renal glucose reabsorption, therefore reducing blood glucose levels in patients with type 2 diabetes. We and others have developed several SGLT2 inhibitors starting from phlorizin, a natural product. Using cryo-electron microscopy, we present the structures of human (h)SGLT2–MAP17 complexed with five natural or synthetic inhibitors. The four synthetic inhibitors (including canagliflozin) bind the transporter in the outward conformations, while phlorizin binds it in the inward conformation. The phlorizin–hSGLT2 interaction exhibits biphasic kinetics, suggesting that phlorizin alternately binds to the extracellular and intracellular sides. The Na$^+$-bound outward-facing and unbound inward-open structures of hSGLT2–MAP17 suggest that the MAP17-associated bundle domain functions as a scaffold, with the hash domain rotating around the Na$^+$-binding site. Thus, Na$^+$ binding stabilizes the outward-facing conformation, and its release promotes state transition to inward-open conformation, exhibiting a role of Na$^+$ in symport mechanism. These results provide structural evidence for the Na$^+$-coupled alternating-access mechanism proposed for the transporter family.

Type 2 diabetes mellitus is characterized by persistent hyperglycemia caused by inadequate insulin action. Chronic high blood sugar damages blood vessels, causing serious health problems such as nephropathy and cardiovascular disease. The primary treatment for diabetes is blood glucose control; drugs that inhibit selective sodium–glucose cotransporter (SGLT1 and SGLT2; also known as SLC5A1 and SLC5A2) (ref. 1) hold great promise for reducing blood glucose levels. Human (h)SGLT1 and hSGLT2 are responsible for reabsorption of glucose in the proximal tubules after filtration through the glomerulus[2]. SGLT2 is located in the S1 and S2 segments of the proximal tubule and absorbs 90% of the filtered glucose; SGLT1, which has higher glucose affinity, is located in the S3 segment of the proximal tubule and absorbs the remaining 10% (ref. 3).

SGLT2, a transmembrane protein with 14 helices (Fig. 1a) and expressed specifically in the kidneys, has 60% sequence homology to SGLT1, which is also expressed in the small intestine. SGLT2 inhibitors that suppress glucose reabsorption and promote urinary excretion are considered a promising therapeutic tool to manage blood glucose levels in patients with type-2 diabetes. SGLT2 inhibitor development initially focused on the natural product phlorizin (Fig. 3b)[4], an O-glucoside that is hydrolyzed by β-glucosidase in the intestine. The poor metabolic stability of phlorizin makes its oral administration difficult. Phlorizin was replaced by N- or C-glucosides, which exhibit better metabolic stability. C-glucosides, including canagliflozin, dapagliflozin and empagliflozin, are marketed as approved drugs in the United States,

[1]Discovery Technology Laboratories Sohyaku Innovative Research Division, Mitsubishi Tanabe Pharma, Yokohama, Japan. [2]DMPK Research Laboratories Sohyaku Innovative Research Division, Mitsubishi Tanabe Pharma, Yokohama, Japan. [3]Department of Biological Sciences, Graduate School of Science, The University of Tokyo, Tokyo, Japan. [4]Present address: Department of Chemistry and Biotechnology, Graduate School of Engineering, The University of Tokyo, Tokyo, Japan. [5]These authors contributed equally: Masahiro Hiraizumi, Tomoya Akashi. ✉e-mail: hiraizumi-masahiro4580@g.ecc.u-tokyo.ac.jp; nureki@bs.s.u-tokyo.ac.jp; miyaguchi.ikuko@mv.mt-pharma.co.jp

Japan and many other countries (Figs. 1b and 2e,f)[5]. SGLT2 inhibitors, generically known as gliflozins, have ushered in a new phase of diabetes treatment, providing many benefits, including reducing the risk of heart failure and improving kidney protection[6–8]. It was initially thought that rare mutations in the Na[+]–glucose cotransporter gene *SLC5A1* can cause lethal glucose–galactose malabsorption and that inhibitors with high specificity for SGLT2 over SGLT1 were necessary to treat diabetes[5]. However, SGLT1 inhibition in the gastrointestinal tract regulates postprandial glucose excursion and gastrointestinal hormone secretion[9]. Therefore, dual inhibitors of SGLT1 and SGLT2 (such as sotagliflozin and LX-2761; Fig. 2c) are currently being developed for the treatment of diabetes[10,11].

SGLT1 and SGLT2 are members of the Sodium-Solute Symporter family of the LeuT transporters, and are conserved in all bacterial and animal taxa, with six isoforms in humans[12,13]. Owing to difficulties in protein preparation and structural analysis, structural homology modeling studies have been conducted using SGLT from *Vibrio parahaemolyticus* (vSGLT)[14,15] and similar protein structures such as LeuT[16] and SiaT[17].

An alternating-access model has been proposed as a functional model of SGLT2 glucose uptake, conserved among LeuT transporters[12]. In this model, the rocking bundle alternately opens outwards and inwards relative to the immobilized scaffold, transporting a substrate molecule per cycle. SGLT2 undergoes a conformational change to an outward-facing conformation by binding to a single Na[+] ion (at the Na2 site, which is conserved among LeuT transporters[13]) before binding to the substrate; this conformational change, which then promotes glucose binding, depends on the Na[+] concentration gradient across the plasma membrane. Subsequently, the Na[+] and sugars are incorporated into the cell in the inward-open conformation[2,12,13]. In contrast, SGLT1 requires two Na[+] ions (at the conserved Na2 and Na3 sites) for glucose transport, potentially affecting its glucose affinity[13].

The structural understanding of the human SGLT family has been advanced by analysis of the cryo-electron microscopy (cryo-EM) structures of SGLT1 and SGLT2. Analysis of the structure of the hSGLT1 apo-form with consensus stabilizing mutations and molecular dynamics calculations has revealed the mechanisms of glucose binding and selectivity and water permeability[18]. The activity of hSGLT2, whose gene has long been difficult to clone, is greatly enhanced by co-expression of MAP17 (PDZK1P1), an essential auxiliary subunit of hSGLT2 (ref. 19). Structural determination of hSGLT2 via MAP17 tethering and the introduction of several mutations has revealed empagliflozin binding in an outward-facing conformation[20]. Furthermore, introduction of MAP17 tethering and several mutations in hSGLT1 has revealed that the dual inhibitor LX2761 binds in the outward conformation of hSGLT1 (ref. 21). The role of residues in SGLT-inhibitor binding in these two outward conformations has been discussed[20,21]. Both moving and constant modules have been defined along with the alternating-access mechanism[21]. Nonetheless, the function of Na[+] remains unknown, because these previous structural studies did not detect Na[+], which should bind to hSGLTs before binding to the substrate. Among the inhibitors, only *C*-glucoside inhibitors have been shown to bind in the outward conformation; the binding mode of *O*- and *N*-glucoside type inhibitors, including phlorizin, remains to be elucidated. Moreover, the conformational change in SGLT2 between the outward and inward conformations and the role of MAP17 in this change remain to be clarified.

In this Article, we performed cryo-EM single-particle analyses to determine the structures of the genuine hSGLT2–MAP17 complexes with five inhibitors (canagliflozin, dapagliflozin, TA-1887, sotagliflozin and phlorizin). Our characterization of the Na[+]-binding outward-facing structures and inward-open structures, together with transport and binding assays, substantially clarifies the molecular features of hSGLT2–MAP17, SGLT2 inhibition and sugar transport. This improves our understanding of the alternating-access mechanism proposed for the transporter family.

## Results

### Structural determination of the hSGLT2–MAP17 heterodimer

We performed cryo-EM analysis of hSGLT2 binding with four glifozins and phlorizin. To obtain a stable and homogeneous sample, we first used fluorescence-detection size-exclusion chromatography (FSEC)[22] to examine the expression of hSGLT2 with its N- or C-terminus fused with enhanced green fluorescent protein (EGFP). However, we were unable to detect hSGLT2 expression (Fig. 1c); given that its N-terminus is exposed to the extracellular side, we hypothesized that N-terminal fusion of a signal sequence and superfolder GFP (sfGFP) would improve the protein expression[23]. Fusing the human trypsinogen 1-derived signal peptide and sfGFP to the hSGLT2 N-terminus revealed a peak, indicating that hSGLT2 can be solubilized by detergents (Fig. 1c). FSEC revealed that co-expression of hMAP17 and hSGLT2 caused a high molecular-weight shift and monodisperse peak formation, suggesting a stable heterodimer-complex formation. hSGLT2–MAP17-expressing cells were able to take up α-methyl-D-glucopyranoside (α-MG; Fig. 1b), reflecting their sensitivity to canagliflozin (Fig. 1d). Analysis of the membrane fraction of hSGLT2–MAP17-expressing cells revealed that the binding activity of hSGLT2 to multiple inhibitors was maintained (Fig. 1e and Extended Data Fig. 1a).

The membrane fraction was solubilized using *N*-dodecyl β-D-maltoside (DDM) micelles in the presence of SGLT2 inhibitors, purified using GFP nanobody-affinity chromatography followed by protease-mediated cleavage of sfGFP, and subjected to gel-filtration column chromatography (Extended Data Fig. 1b,c). Cryo-EM single-particle analyses were applied to the purified hSGLT2–MAP17 complexes with the five inhibitors (Supplementary Figs. 2–6). The acquired movies were motion-corrected and processed in RELION[24,25], providing cryo-EM maps at overall resolutions of 2.6–3.3 Å, according to the gold-standard Fourier shell correlation (FSC) 0.143 criterion (Table 1). All of the potential maps contain disordered regions but are sufficient for building structural models of the proteins and inhibitors (Extended Data Figs. 2 and 4). The overall structure exhibits a LeuT fold comprising 14 membrane-spanning helices (transmembrane domain (TM)0–13). Helices TM1–5 and TM6–10 form an inverted repeating structure (Fig. 1a). *N*-glycan, attached to N250 of hSGLT2, was identified (Extended Data Figs. 2 and 4). Canagliflozin, dapagliflozin, sotagliflozin and TA-1887 were found to bind to the outward-facing conformations, with Na[+] bound to the conserved Na2 sites, whereas phlorizin was found to bind to the inward-open conformation without Na[+] binding. Because of their high flexibility, the N-terminal loop of hSGLT2 (15–20 amino acids), the IL6 loop between TM12 and TM13, and the extracellular or intracellular regions of MAP17 were not visible in either structure. Furthermore, the inward-open structure does not exhibit density for IL0 between TM0 and TM1. In MAP17, a single transmembrane helix interacts with TM13 of hSGLT2. The extracellular half of the single helix interacts closely with hydrophobic residues and lipids, while the intracellular half does not interact with hSGLT2 (Fig. 1f and Supplementary Fig. 8).

### Comparison of hSGLT2 structures in outward-facing conformation with gliflozins

The *C*-glucoside inhibitors (canagliflozin, dapagliflozin and sotagliflozin), and the *N*-glucoside TA-1887 were bound to the central hydrophobic cavity of the topologically inverted repeats (IRs) (TM1, TM2, TM3, TM6 and TM10; Figs. 1a and 2a) of hSGLT2 in the outward-facing conformation. This cavity was negatively charged, favoring the binding of positively charged ions such as Na[+].

The binding mode that we observed was consistent with that revealed in a recent study reporting the SGLT2–empagliflozin structure without confirming Na[+] binding[20]. When the overall structures (including those of MAP17) are superimposed, they are almost identical (root-mean-square deviation (RMSD) 0.66–0.85 Å) (Supplementary Fig. 7). The only difference is in the loop connecting

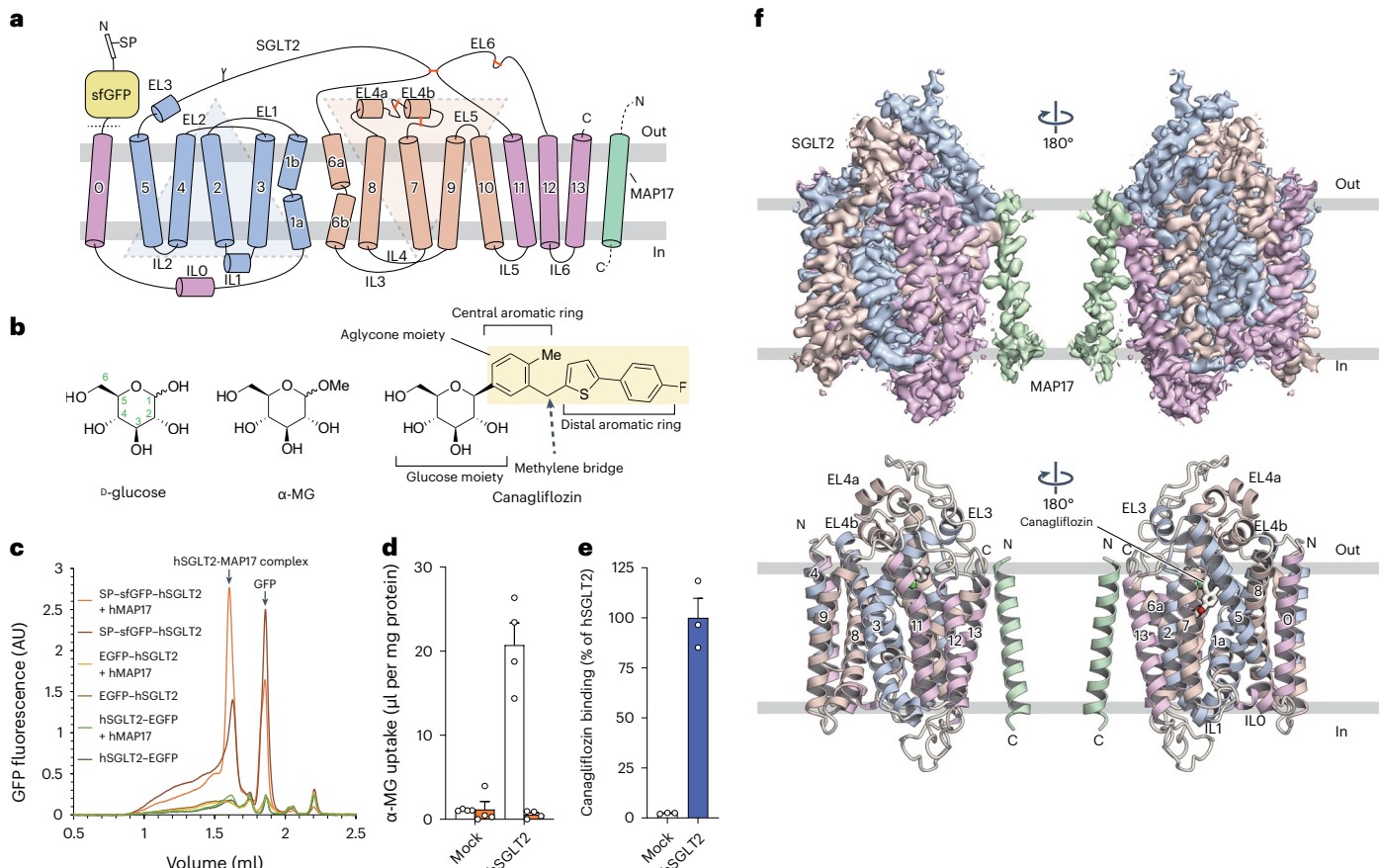

**Fig. 1 | Biochemical and cryo-EM studies of the human SGLT2–MAP17 heterodimer. a**, Topological diagram of the hSGLT2–MAP17 heterodimer. IRs are represented as light blue and light brown. The Y shape indicates an N-glycosylation site. Disulfide bonds are shown as orange sticks. A signal peptide (SP) and superfolder (sf) GFP were fused to the N-terminus of hSGLT2. The dotted line indicates the site of protease activity. **b**, Chemical structures of the substrates and a representative gliflozin. **c**, FSEC profiles of various types of GFP-tagged hSGLT2 in the presence and absence of MAP17. Arrows indicate elution positions of the hSGLT2–MAP17 heterodimer and free GFP (GFP). **d**, α-MG uptake in hSGLT2- and MAP17-expressing cells in the absence (white) or presence (orange) of 500 nM canagliflozin. Each column represents mean ± s.e.m. ($n$ = 4, biological replicates). **e**, Canagliflozin (30 nM) binding assay for the hSGLT2- and MAP17-expressing cell membrane fraction. Each column represents mean ± s.e.m. ($n$ = 3, technical replicates). **f**, Overall structure of the human SGLT2–MAP17 complex. Cryo-EM maps (top) and ribbon models (bottom). The same color scheme was used throughout the manuscript, except for Fig. 5.

TM12 and TM13: in the structure of Niu et al.[20], GFP is fused, whereas in our study, this region was not well visualized, owing to the flexibility of the structure. Upon comparing these studies, we found no differences in the main or side chain structures, although there were differences in the presence or absence of Na⁺ binding and in MAP17 tethering. We believe that there are no differences that warrant consideration when discussing the binding modes of the inhibitors (Fig. 2b–f).

Most SGLT2 inhibitors were composed of glucose and aglycone moieties (Fig. 1b). These four gliflozins have similar IC₅₀ values (1–6 nM) and similar binding modes (Fig. 2b–f). The aglycone moiety comprises two aromatic rings that bend at the methylene bridge and extend toward the extracellular space (Fig. 2a,b). The glucose moiety of all gliflozin inhibitors stacks with the aromatic side chain of the inner gate, Y290. The hydroxyl groups of the glucose moiety form hydrogen bonds in the side chains of N75, H80, E99, S287, W291, K321 and Q457 and in the main chain carbonyl group of F98 (Fig. 2b); these residues are conserved in SGLT1 (Supplementary Fig. 1). A water molecule-like density was observed near S460 in the structures of canagliflozin, sotagliflozin, dapagliflozin and TA-1887 (Fig. 2b–e and Extended Data Fig. 2). Based on water analysis using 3D-RISM, these S460 and water-mediated hydrogen bonds appear to be energetically unstable and contribute little to the affinity (Extended Data Fig. 3a–c).

## Subtype specificity between SGLT1 and SGLT2

There are strong similarities between SGLT1 and SGLT2 in both sequence and functionality, although they differ in the number of Na-binding sites. SGLT1 has about fivefold higher substrate affinity than SGLT2, whereas SGLT2 has higher sugar-transport capacity and selectivity for phlorizin[3,26]. Canagliflozin, dapagliflozin, TA-1887 and empagliflozin exhibit SGLT2-specific inhibition, whereas LX-2761 (reported in complex with hSGLT1) and sotagliflozin are dual inhibitors against SGLT1 and SGLT2 (refs. 11,21). We next investigated the subtype specificity of these SGLT inhibitors.

Comparing the outward-facing structures of hSGLT1 reported by Niu et al.[21] and of the four SGLT2-binding gliflozins in this study, the RMSD of the main chain ranged from 1.09 Å to 1.23 Å, and the overall structures were almost identical (Supplementary Fig. 7). In the sugar moiety, the differences in residues between SGLT1 and SGLT2 are concentrated around C6-OH of the sugar, forming hydrophobic pockets involving S460, V286 and L283 in SGLT2, and T460, L286 and M283 in SGLT1. This explains why this hydrophobic pocket is larger for SGLT2 (Extended Data Fig. 3d–f)[21].

Dapagliflozin and sotagliflozin differ in the C₆ functional group, exhibiting -OH and -SCH₃, respectively. This indicates that, although the important hydrogen bond with Q457 is lost, their activity against SGLT2 is maintained owing to the hydrophobic interaction resulting

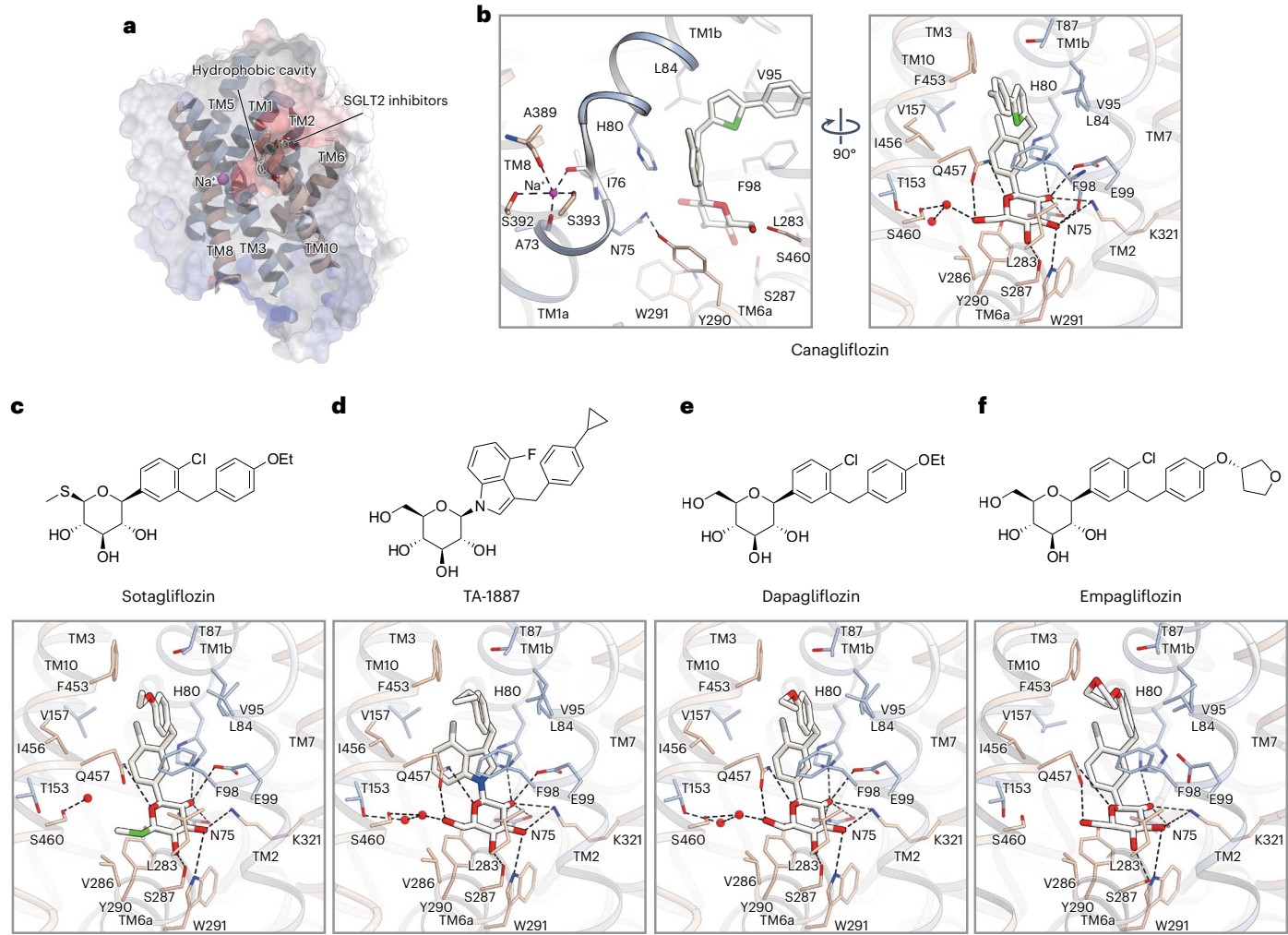

**Fig. 2 | SGLT2-inhibitor-binding site of the outward-facing conformation.**
**a**, Cross-sections of the electrostatic surface potentials at the SGLT2 inhibitor-binding site. The potentials are displayed as a color gradient from red (negative) to blue (positive). **b**, Interactions between Na$^+$ ions (purple spheres), canagliflozin and hSGLT2. Canagliflozin and its interacting residues are shown as sticks. Hydrogen bonds are indicated by black dashed lines. **c–f**, Chemical structures of inhibitors and their interactions with hSGLT2: sotagliflozin (**c**), TA-1887 (**d**) and dapagliflozin (**e**) in this study, and of empagliflozin (**f**) (previously reported[20]).

from the displacement of an unstable water molecule (Extended Data Fig. 3a–c). In SGLT1, this pocket is smaller, and the effect of the hydrophobic interaction is stronger, explaining why sotagliflozin can function as a dual inhibitor. Although dapagliflozin differs from sotagliflozin only in its $C_6$-OH group, there is in fact a more than 1,000-fold difference in inhibition between SGLT1 and SGLT2 (ref. 5), which cannot be explained by the sequence difference at the sugar binding site alone. D268, in the extracellular loop EL5, reportedly influences subtype selectivity, even though it is not directly involved in the binding pocket[26]. Conformational changes in SGLTs between the inward- and outward-open forms rely on the membrane potential and Na$^+$ binding, via rearrangements among the TMs that constitute the substrate-binding site. Factors other than direct binding may slightly alter the structure and stability of the substrate-binding site, potentially contributing to the subtype specificity of dapagliflozin.

Besides, variation in the aglycon moiety contributed to SGLT-subtype selectivity. In all of the gliflozin inhibitors, the central aromatic ring is exposed to the hydrophobic cavity formed by TM1, TM3 and TM10 (Fig. 2). Specifically, TA-1887 comprises a benzylindole ring, which extends into the hydrophobic cavity (Fig. 2d). The hydrophobic substituents positioned at the *para* position influence the inhibitory

activity of hSGLT2, and are all located in roughly the same region (Fig. 2b–f). This site is potentially selective for hSGLT2, given the size differences in the hydrophobic pockets. Previous analysis of molecular dynamics simulation of mizagliflozin–hSGLT1 binding revealed that A160, corresponding to V157 of hSGLT2, affects this selectivity[21], consistent with our current findings.

Distal aromatic rings form a long hydrophobic aglycon tail that extends into the extracellular vestibule (Fig. 2b–f). In all of the gliflozins, each tail forms T-shaped π–π stacking with F98 of TM2, and is surrounded by hydrophobic residues including L84 of TM1, V95 of TM2, and F453 of TM10. Canagliflozin extends with fluorophenyl via a thiophene ring, forming a hydrophobic interaction with the extracellular vestibule (Fig. 2b). Structure–activity relationship studies during the development of canagliflozin revealed that its inhibitory activity increases with furan replaced by thiophene in the center[27], suggesting that this moiety should be hydrophobic. Therefore, these hydrophobic residues play important roles in the inhibitory action. Furthermore, SGLT1 has I98 at the position corresponding to V95 of TM2 in hSGLT2, and V95I reduces the inhibitory activity of empagliflozin in hSGLT2 (ref. 20). The distal aromatic rings contribute to the selectivity between hSGLT1 and hSGLT2 (ref. 8).

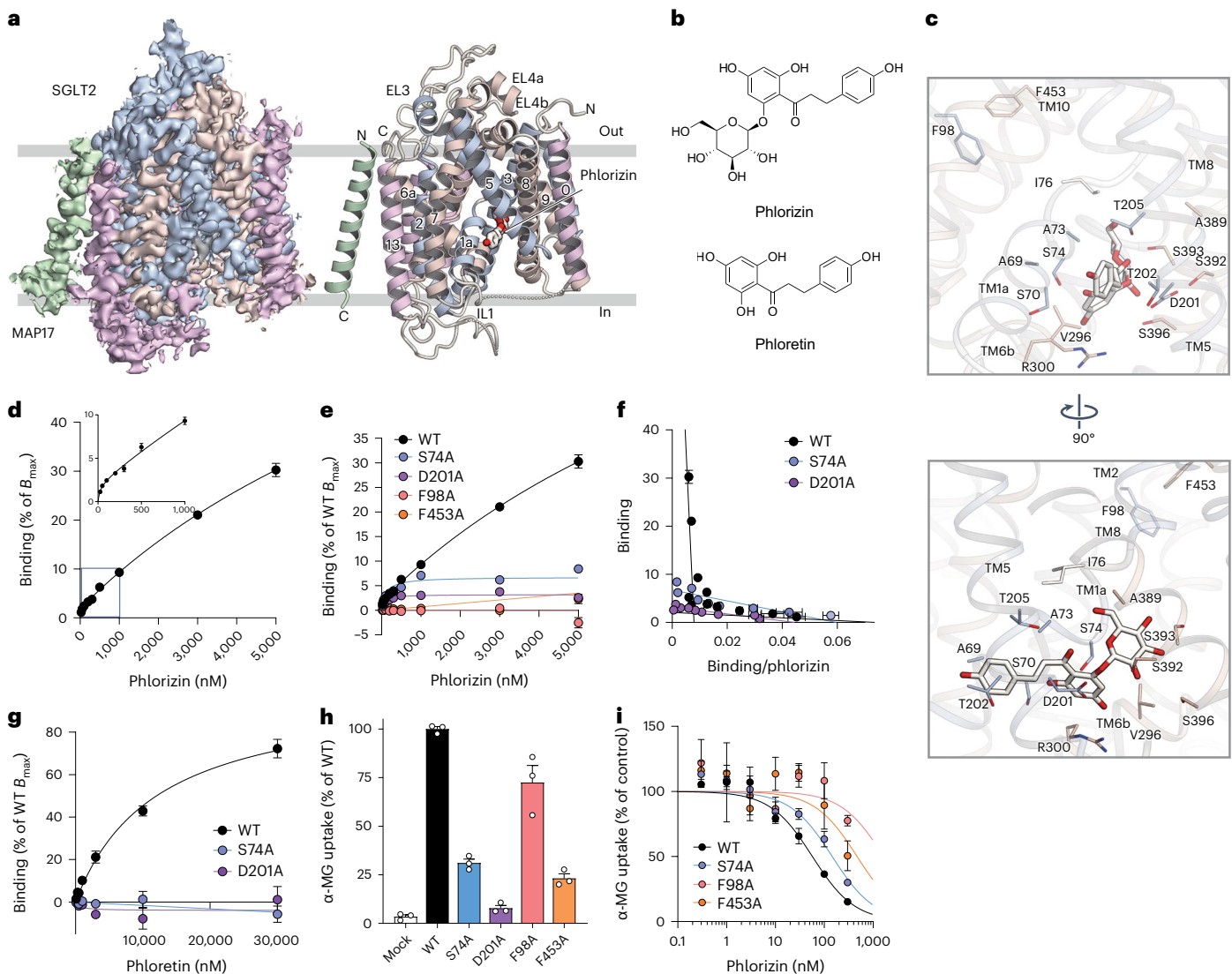

**Fig. 3 | Phlorizin-binding site of the inward-open conformation. a**, Inward-open conformation of the hSGLT2–MAP17 complex. Cryo-EM maps (left) and ribbon models (right). **b**, Chemical structures of phlorizin and phloretin. **c**, Interactions between phlorizin and hSGLT2. **d**, Concentration-dependent binding of phlorizin to the WT hSGLT2. Each point represents the mean ± s.e.m. ($n = 3$, technical replicates). **e**, Concentration-dependent binding of phlorizin in WT and mutant hSGLT2. Each point represents the mean ± s.e.m. ($n = 3$, technical replicates).

**f**, Eadie–Hofstee plot analysis of phlorizin binding in WT and mutant hSGLT2. Each point represents the mean ± s.e.m. ($n = 3$, technical replicates). **g**, Concentration-dependent binding of phloretin in WT and mutant hSGLT2. Each point represents the mean ± s.e.m. ($n = 4$, technical replicates). **h**, Uptake assay of α-MG in WT and mutant hSGLT2. Each column represents mean ± s.e.m. ($n = 3$, biological replicates). **i**, Inhibitory effect of phlorizin on α-MG uptake by WT and mutant hSGLT2. Each point represents the mean ± s.e.m. ($n = 3$, biological replicates).

## High-concentration phlorizin fixes hSGLT2 in the inward-open state

Unexpectedly, phlorizin was found to bind to TM1, TM5 and TM8 in an inward-open structure (Fig. 3a,b and Extended Data Fig. 4), in contrast to the outward-facing structure that we observed for the gliflozins. This binding site is located near the Na2 site, where Na⁺ binds to the outward-facing structure (Fig. 2b). The glucose moiety of phlorizin is bound to the bending site of the intracellular side of TM1, whereas the aglycon moiety, connected to the glucose moiety via an ether bond, is surrounded by the side chain comprising A69, S70, A73 and S74 of TM1, D201 of TM5, and R300 of TM6, extending toward the lipid membrane (Fig. 3c). The ether bond is unique to phlorizin, whereas the central aromatic ring of the gliflozins connects directly to the glucose moiety, causing rigidity that prevents binding to the inward-open structure.

Phlorizin at up to 5,000 nM did not saturate the hSGLT2-expressing membrane; it exhibited biphasic kinetics, indicating that phlorizin has

two binding sites on hSGLT2 (high- and low-affinity sites; Fig. 3d–f). Similarly, [³H]phlorizin exhibits biphasic binding in the renal plasma membrane in rats[28]. In whole-cell clamp experiments, hSGLT2 inhibitors, including phlorizin, achieve inhibition by acting on the extracellular side[29]. However, among the SGLT2 inhibitors, only phlorizin has been reported to act weakly from the intracellular side at high concentrations in the absence of Na⁺ gradient[29]. In the phlorizin-bound inward-open structure, the extracellular-binding site is abolished, suggesting that phlorizin can bind to only one of the sites (extracellular or intracellular). Under the conditions of our cryo-EM analysis (no Na⁺ gradient and 500 μM phlorizin), phlorizin bound to the low-affinity intracellular site (Fig. 3d). We therefore believe that our cryo-EM analysis revealed binding from the intracellular side.

To confirm the binding sites, we examined the binding activity and transport functions of hSGLT2 alanine mutants of residues S74 or D201 in phlorizin binding in the inward-open structure and of F98 or

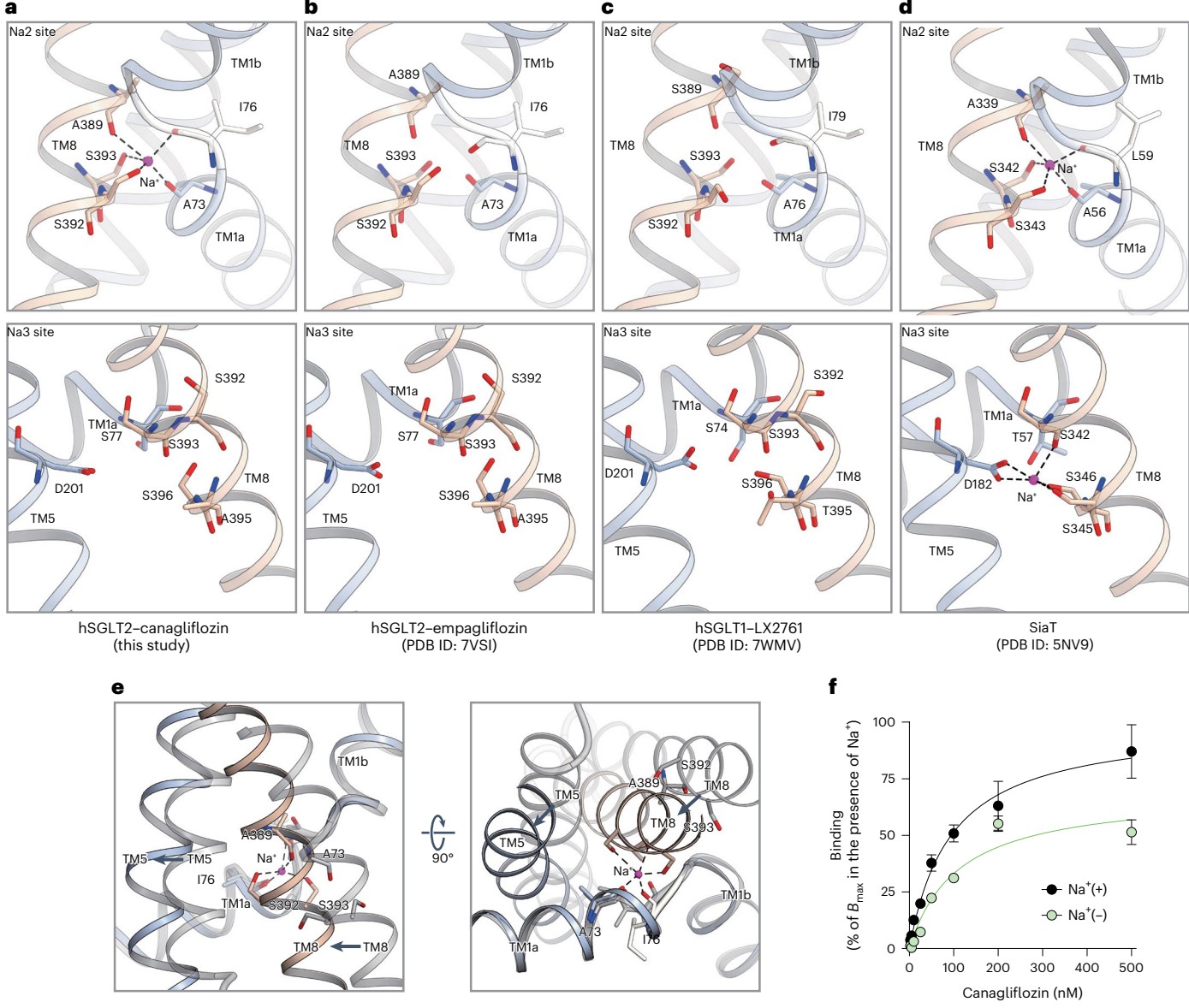

**Fig. 4 | Na⁺-binding sites of SGLT1 and SGLT2. a–d,** Comparison of conserved Na+-binding sites in Sodium-Solute Symporter family. Canagliflozin bound hSGLT2 in this study (**a**), empagliflozin bound hSGLT2 (PDB ID : 7VSI) (**b**), LX2761 bound hSGLT1 (PDB ID : 7WMV) (**c**) and Proteus mirabilis sialic-acid transporter (SiaT) (PDB ID : 5NV9) (**d**). Residues that coordinate Na⁺ ions (purple spheres) are depicted. The Na2 and Na3 sites are displayed above and below, respectively. Na⁺ binding is not observed for SGLT2–empagliflozin and SGLT1–X2761 at the Na2 site. Na⁺ binding at the Na3 site is only observed in SiaT. Notably, T395

coordinates with Na⁺ in SGLT1, whereas A395 in SGLT2 renders Na⁺ incapable of binding. The relative positions of the residues are similar. **e,** Conformation change from outward-facing (canagliflozin-bound structure, color) to inward-open (phlorizin-bound structure, gray) around the Na2 binding site. View from the lateral side of the plasma membrane (left) and from the intracellular side (right). **f,** Concentration-dependent binding of canagliflozin in the presence and absence of Na⁺. The specific binding was normalized to the $B_{max}$ in the presence of Na⁺. Each point represents the mean ± s.e.m. ($n$ = 3, technical replicates).

F453 in inhibitor interactions in the outward-facing structure. Based on FSEC, both the mutants and wild type (WT) preserved their conformation (Extended Data Fig. 5). Unexpectedly, the F98A and F453A single mutants did not bind to phlorizin (Fig. 3e). F98 and F453 not only participate in inhibitor binding in the outward-facing conformation but also form π–π stacking interactions with each other in the inward-open conformation (Fig. 3c).

The low-affinity binding phase of phlorizin was lost in both the S74A and D201A mutants, with phlorizin binding only to the high-affinity site of the WT (Fig. 3e,f and Extended Data Table 1). Phloretin, the aglycon tail of phlorizin, binds as effectively as phlorizin to the WT but not to the S74A and D201A mutants (Fig. 3g). Therefore, S74A and D201A have lost the ability to bind phlorizin on the

intracellular side but can still bind it extracellularly. Since the S74A, F98A, and F453A mutants maintained α-MG uptake (Fig. 3h), we performed experiments to inhibit sugar uptake using phlorizin. Inhibition of α-MG uptake by phlorizin was greatly impaired in the F98A and F453A mutants, which lacked phlorizin-binding ability but was maintained in the S74A mutant, in which the extracellular binding site was functional (Fig. 3i and Extended Data Table 2). No clear uptake of α-MG was observed in the D201A mutant (Fig. 3h). In hSGLT1, D201 corresponds to D204, which is involved in Na3-site formation and is important for sugar-uptake activity and cell trafficking; it is expected to play a similar role in hSGLT2 (ref. 30). Our results support the earlier suggestions that phlorizin inhibits SGLT2 strongly from the extracellular side and weakly from the intracellular side[29].

## Table 1 | Cryo-EM data collection, refinement and validation statistics

| | Canagliflozin (EMDB-34673), (PDB 8HDH) | Dapagliflozin (EMDB-34705), (PDB 8HEZ) | TA-1887 (EMDB-34610), (PDB 8HBO) | Sotagliflozin (EMDB-34737), (PDB 8HG7) | Phlorizin (EMDB-34823), (PDB 8HIN) |
|---|---|---|---|---|---|
| **Data collection and processing** | | | | | |
| Magnification | 215,000 | 105,000 | 105,000 | 105,000 | 105,000 |
| Voltage (kV) | 300 | 300 | 300 | 300 | 300 |
| Electron exposure (e⁻ Å⁻²) | 64 | 64 | 64 | 64 | 64 |
| Defocus range (μm) | −0.6 to −1.6 | −0.8 to −1.6 | −0.8 to −1.6 | −0.8 to −1.6 | −0.8 to −1.6 |
| Pixel size (Å) | 0.4 | 0.83 | 0.83 | 0.83 | 0.83 |
| Symmetry imposed | C1 | C1 | C1 | C1 | C1 |
| Initial particle images (no.) | 2,364,108 | 3,692,950 | 3,395,470 | 5,242,427 | 3,013,029 |
| Final particle images (no.) | 65,919 | 197,695 | 103,853 | 72,773 | 76,485 |
| Map resolution (Å) | 3.1 | 2.8 | 2.9 | 3.1 | 3.3 |
| FSC threshold | 0.143 | 0.143 | 0.143 | 0.143 | 0.143 |
| Map resolution range (Å) | 3.2–4.8 | 2.4–5.2 | 2.6–5.4 | 2.7–6.5 | 2.6–3.1 |
| **Refinement** | | | | | |
| Initial model used (PDB code) | | | | | |
| Model resolution (Å) | 3.1 | 2.6 | 2.7 | 3.0 | 0.5 |
| FSC threshold | 0.5 | 0.5 | 0.5 | 0.5 | 3.3 |
| Map sharpening $B$ factor (Å²) | −107.9 | −95.4 | −75.8 | −96.4 | −144.5 |
| Model composition | | | | | |
| Non-hydrogen atoms | 4,728 | 4,692 | 4,728 | 4,763 | 4,743 |
| Protein residues | 1 | 1 | 1 | 1 | 0 |
| Ligands | 49 | 46 | 49 | 44 | 45 |
| $B$ factors (Å²) | | | | | |
| Protein | 98.6 | 83.9 | 70.9 | 75.0 | 123.3 |
| Ligand | 131.0 | 126.6 | 113.0 | 128.2 | 209.0 |
| RMSDs | | | | | |
| Bond lengths (Å) | 0.013 | 0.015 | 0.013 | 0.014 | 0.018 |
| Bond angles (°) | 2.013 | 2.241 | 2.017 | 2.015 | 2.368 |
| Validation | | | | | |
| MolProbity score | 1.7 | 1.79 | 1.63 | 1.86 | 2.37 |
| Clashscore | 5.21 | 5.45 | 5.93 | 6.10 | 8.30 |
| Poor rotamers (%) | 1.40 | 2.82 | 0.80 | 1.99 | 4.18 |
| Ramachandran plot | | | | | |
| Favored (%) | 95.6 | 97.19 | 95.6 | 95.8 | 92.9 |
| Allowed (%) | 3.6 | 2.64 | 4.1 | 4.2 | 6.9 |
| Disallowed (%) | 0.8 | 0.2 | 0.3 | 0.0 | 0.2 |

### Role of Na⁺ in SGLT1/2

The Na⁺-binding Na2 site, which is conserved in many LeuT fold transporters, is located near the middle bend of TM1 (Figs. 2b and 4a). In SGLT2, the Na2 site is thought to be formed by the backbone carbonyls of A73, I76 and A389 and the side-chain oxygens of S392 and S393 (Figs. 2b and 4a), based on the alignment of vSGLT[15] and SiaT[17] (Fig. 4a–d and Supplementary Fig. 1). The density corresponding to Na⁺ at the Na2 site was not observed in hSGLT2–empagliflozin[20] or hSGLT1–LX2761 (ref. 21), but was confirmed in all four of our outward-facing structures (Extended Data Fig. 6). As predicted from the alignment, Na⁺ interacts with A73 and I76 of TM1 and with A389, S392 and S393 of TM8, resulting in a trigonal bipyramidal form (Fig. 4a). Therefore, TM1 and TM8 are connected by Na⁺ and form part of the substrate-binding site (Fig. 4e).

In the outward-facing structure, Na⁺ not only connects TM1 and TM8 but also enables the entire outer region of the glucose-binding site, including TM5, to move outward, causing the change from inward-open to outward-facing (Fig. 4e). It has been suggested that all conformations of vSGLT are in dynamic equilibrium[31]. In the presence of Na⁺, vSGLT favors the outward-facing conformation, while this conformation occurs less frequently in the absence of Na⁺. Here, for canagliflozin and TA-1887, the maximum number of binding sites ($B_{max}$) was partially lower in the absence than in the presence of Na⁺ (Fig. 4f, Extended Data Fig. 6e,f and Extended Data Table 3). These results indicate that Na⁺ is involved in the stabilization of the outward conformation in SGLT2, as reported in SGLT1 (ref. 32). However, as Na⁺ and sugar uptake occurs at a 1:1 ratio for SGLT2 (ref. 2), it is likely that sugar uptake and the associated

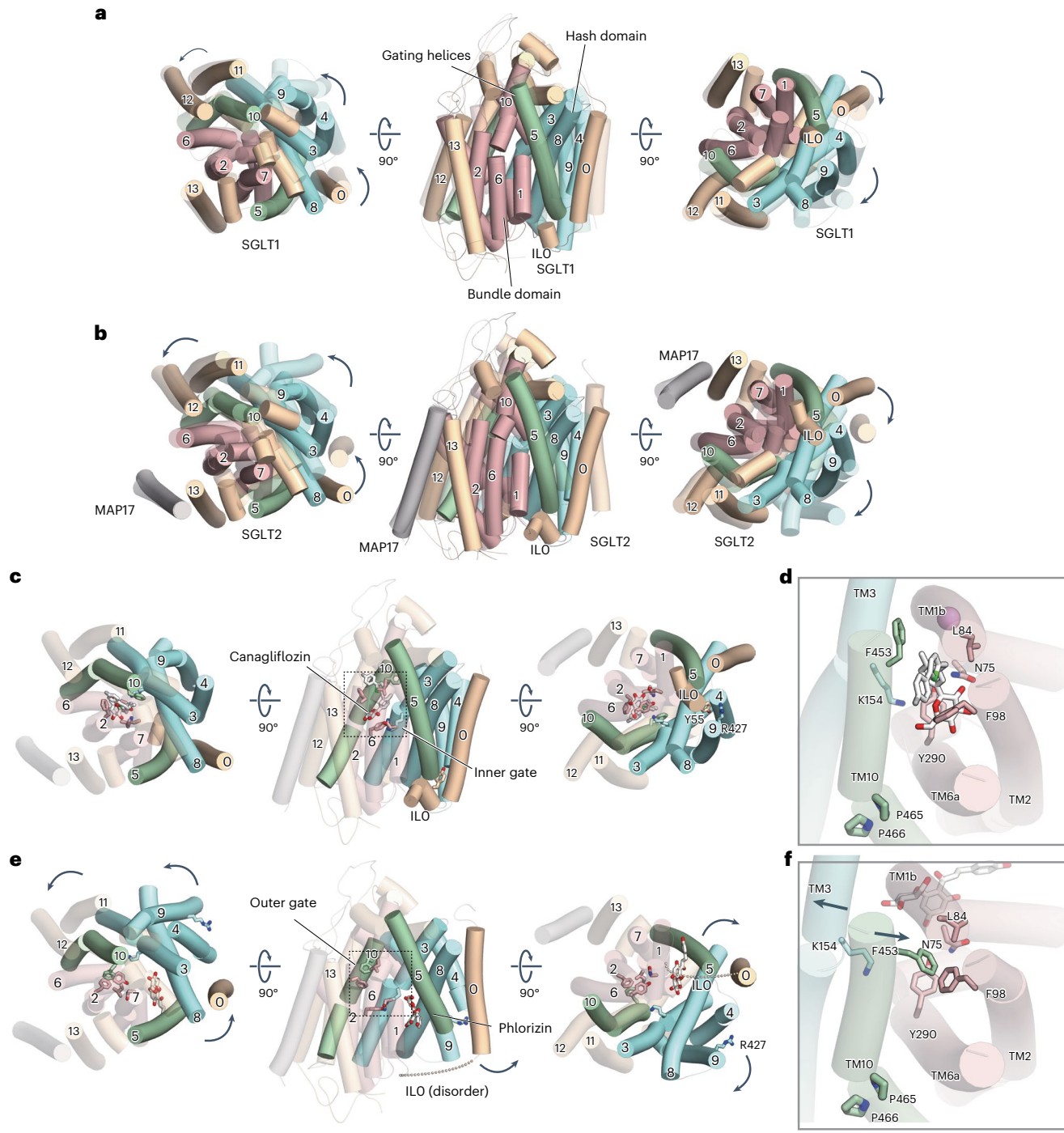

**Fig. 5 | Comparison of the outward-facing and inward-open conformations of hSGLT1/2. a**, Superposition of hSGLT1 outward-facing (colored, PDB ID: 7WMV) and inward-open (transparent, PDB ID: 7SLA). **b**, Superposition of hSGLT2 outward-facing (colored) and inward-open (transparent). Outward-facing and inward-open structures when their bundle domains (TM1, TM2, TM6 and TM7; red) are superimposed. MAP17 (gray) and TM13 (light orange) overlap well between the conformations, but the hash domain (TM3, TM4, TM8 and TM9; blue), gate helices (TM5 and TM10; green), TM0, TM11 and TM12 (light orange) are in the inward-open conformation. **c**, The outward-facing conformation of the hSGLT2–MAP17 complex with canagliflozin viewed from the exoplasm (left), side (center) and cytoplasm (right). **d**, Substrate sugar-binding site and external vestibule of the outward-facing conformation from the exoplasm. **e**, The inward-open conformation of the hSGLT2–MAP17 complex with phlorizin, viewed from the exoplasm (left), side (center) and cytoplasm (right). **f**, Substrate sugar-binding site and external vestibule of the inward-open conformation from the exoplasm.

conformational change to the inward structure are only possible when Na[+] is bound. This is consistent with the fact that, in SGLT1, substitutions of S392 and S393 at the Na2 site reduced glucose uptake[33].

SGLT1 requires two Na[+] ions for sugar transport and has both Na2 and Na3 sites[2]. In SGLT2, the region corresponding to the SGLT1 Na3 site, located on the cytoplasmic side and away from the glucose-binding pocket, is occluded by the side-chain carboxyl group of D201 in TM5 and the backbone carbonyls of S392, S393, A395 and S396 in TM8 (Fig. 4a–c, Extended Data Fig. 6a–d and Supplementary Fig. 1). In SGLT1, the residue corresponding to A395 in hSGLT2 is

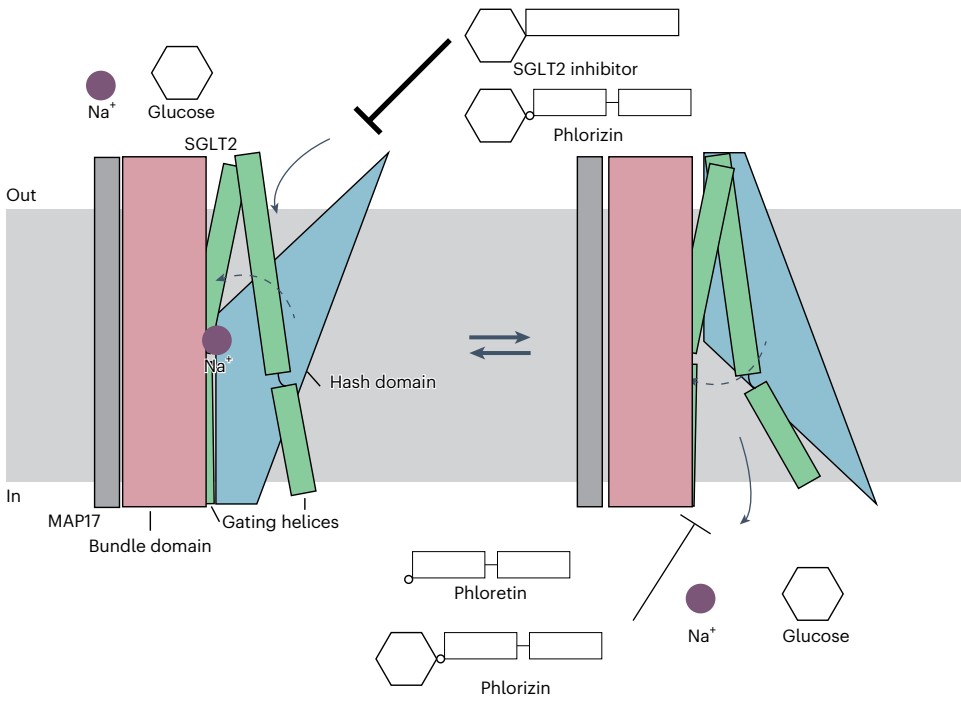

**Fig. 6 | Proposed SGLT2 transport and inhibition mechanism.** The bundle domain is anchored to the membrane with MAP17, and the rest of the transporter undergoes a conformational change according to an alternating-access mechanism. Upon sodium binding, the transporter opens outward to allow a substrate or inhibitor to bind. Upon substrate-binding, the inner gate opens and sodium and glucose are released into the cell. After sodium and glucose are released, the transporter forms an inward-open structure and phlorizin and phloretin bind to this structure, inhibiting glucose transport.

replaced by Thr, which probably contributes to Na⁺ binding (Fig. 4c). This is consistent with the fact that no density corresponding to Na⁺ was observed at the Na3 site in the hSGLT2 structures (Fig. 4a,b and Extended Data Fig. 6b).

The current outward-facing conformation of hSGLT2 exhibits a region that closely resembles the SGLT1 Na3 site and partially shares the same residues as the Na2 site. Despite the lack of the Na3 site in SGLT2, the positions of the residues at the Na3 site are identical in SGLT2 and SGLT1, as is the position of TM5, which connects to TM1 via Na3 (Fig. 4−c). In SGLT1, replacement of T395 with Ala at the Na3 site resulted in a more than 30-fold reduction in glucose affinity[26]. This indicates that Na3 is associated with glucose transport and may be closely associated with differences in functional activity between SGLT1 and SGLT2.

## Discussion

We elucidated the structures of five different hSGLT2−MAP17−inhibitor complexes and described the sodium-binding outward-facing and inward-open structures of hSGLT2, as well as its biphasic inhibition by phlorizin. We now describe the alternating-access mechanism underlying structural rearrangement and discuss our results.

In SGLT2, the constant immobile module (TM1, TM2, TM6, TM7, TM11, TM12 and TM13) containing the bundle domain (TM1, TM2, TM6 and TM7) is fixed as a scaffold; the moving module (TM0, TM3, TM4, TM5, TM8, TM9 and TM10), containing the hash domain (TM3, TM4, TM7 and TM8) and the gating helix (TM5 and TM10), determines the conformation (inward-open versus outward-facing)[12,21].

In the inward-facing conformation, reported for vSGLT[14,15] and hSGLT1 (ref. 18), the size of the groove that opens inwardly differs depending on the binding ligands (Supplementary Fig. 9). Furthermore, the movement of the TM regions of SGLT1 and hSGLT2, which form the moving module, are similar, indicating that they mimic the structural changes that occur during sugar transport (Fig. 5a,b). Given that hSGLT2 exhibits outward and inward conformations, we next review the overall structural rearrangement of TMs according to the alternating-access mechanism shared by the LeuT transporters[12].

In addition to SGLT2's TMs, the transport function of SGLT2 requires the single TM of MAP17 as an auxiliary subunit. When the bundle domain is superimposed between the inward and outward structures, MAP17 is also well superimposed, while the moving module of the transporter alters its position and conformation (Fig. 5b and Supplementary Video 1). MAP17 co-expression enhances hSGLT2 activity without altering hSGLT2 expression on the cell membrane[19]. However, recent surface labeling experiments using antibodies against the extracellular domain of hSGLT2 have demonstrated that MAP17 is essential for robust surface hSGLT2 expression[34]. Although MAP17 is anticipated to induce a structural change in the active state of hSGLT2 (ref. 19), the current structures argue against MAP17 directly affecting hSGLT2 activity. The decrease in thermal stability observed in the absence of MAP17 suggests that MAP17 contributes by optimizing the localization and folding of SGLT2 in the plasma membrane, facilitating glucose reabsorption (Extended Data Fig. 7). In summary, MAP17, together with the bundle domain, is considered a part of the fixed scaffold that provides alternating access to SGLT2 and contributes to its stabilization.

SGLT2 also has gating functions required for alternating access. The substrate-binding site and external vestibule are formed by TM1, TM2, TM6 and TM10 in the outward-facing conformation (Fig. 5c,d). Y290 in TM6, N75 in TM1, and K154 in TM3 form π−cation interactions, where the corresponding interaction of vSGLT1 is thought to act as an inner gate for substrates. SGLT1/2 have a characteristic Pro−Pro motif (465, 466) in the middle of TM10, causing a bend in the α-helix (Fig. 5e,f). This causes F453 of TM10 to form a T-shaped π−π interaction with F98 of TM2 in the inward-open conformation; the external vestibule is covered by this interaction and by L84 of TM1 (Fig. 5f). In alternating access, there should be no leakage from outside the cell when the transporter is in an inward conformation[35]. These residues are thought to act as external gates for transporters. The binding of the distal aromatic ring to the outward-facing

conformation moves the F453 side chain of TM10 to the opposite side; instead, the distal aromatic ring forms a tight interaction with F98 of TM2, thus inhibiting the inward transition (Supplementary Video 1). This is consistent with the reduced binding activity of SGLT2 inhibitors in the F98A and F453A mutants (Extended Data Fig. 5b).

Here we investigated the conformational change from outward to inward structure. With the movement of the hash domain and gating helices, the interaction between K154 of TM3 and Y290 of TM6 is broken (Fig. 4d,f), and TM8 moves so that it fills the space partially occupied by TM3. Furthermore, TM8 and the intracellular part of TM9 shift outwardly. The small-loop structure of IL0 stabilizes the outward-facing structure from the intracellular side, together with a cation–π interaction between R427 and Y55 that becomes lost and disordered in the inward-open conformation. In SGLT1 and vSGLT, the small helical structure IL0 in the inward-open conformation is also disordered[14,18], and is therefore thought to be conserved among these proteins.

Although this study does not fully elucidate the dynamics of $Na^+$ binding/release and sugar uptake, it reveals that sugar uptake depends on the change from the outward-facing to the inward-open conformation. Furthermore, groove formation by the inward-open structure without $Na^+$ binding is the driving force of sugar uptake to cytoplasm. Based on molecular dynamics studies of vSGLT, sugar uptake occurs after $Na^+$ is released[14], consistent with our hypothesis. After the uptake of sugar and sodium is completed in the inward-open state, the sodium concentration gradient and membrane potential[31] drive the change toward the outward-open conformation, allowing the next sodium ion to be accepted, thus rotating the glucose transport cycle. Our structural findings therefore provide support for the proposed $Na^+$–glucose co-transport mechanism (Fig. 6).

Most of the Sodium-Solute Symporter family probably share common transport mechanisms with SGLT1 and SGLT2, as they have common Na2 sites and domain architectures. We believe that these findings will help to improve our understanding of the molecular regulation of this transporter family and to develop new drugs targeting disease-related transporters.

## Online content

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

## Methods

### Reagent and chemicals

Canagliflozin and TA-1887 were synthesized by Mitsubishi Tanabe Pharma Corporation[27,36]. Dapagliflozin and sotagliflozin were purchased from Cayman Chemical Company. Phlorizin, phloretin and α-MG were purchased from Sigma-Aldrich.

### cDNA constructs

hSGLT2 complementary DNA and human MAP17 cDNA were synthesized and codon-optimized for expression in human cell lines. Both cDNAs were cloned into the pcDNA3.4 vector. The hSGLT2 sequence was fused with an N-terminal signal sequence from human trypsinogen 1, a His10 tag, and sfGFP, followed by a human rhinovirus 3C protease (HRV3C protease) recognition site. Point mutations were introduced into this construct using site-directed mutagenesis. These plasmids were utilized for all experimental procedures conducted in this study.

### Expression and purification of the hSGLT2–MAP17 heterodimer

Mammalian Expi293 cells (Thermo Fisher Scientific) were grown and maintained in Expi293 Expression Medium at 37 °C and 8% $CO_2$ under humidified conditions. Cells were transiently transfected at a density of $2.0 \times 10^6$ cells ml$^{-1}$ with the plasmids and FectoPRO (Polyplus). Approximately 320 μg of the hSGLT2 plasmid and 160 μg of the MAP17 plasmid were premixed with 720 μl of FectoPRO reagent in 60 ml of Opti-MEM (Gibco, Thermo Fisher Scientific) for 10–20 min before transfection. For transfection, 60 ml of the mixture was added to 0.6 liters of the cell culture and incubated at 37 °C in the presence of 8% $CO_2$ for 72 h before collection. The cells were collected via centrifugation (800$g$, 10 min, 4 °C) and stored at −80 °C before use. The detergent-solubilized proteins were analyzed via FSEC using an ACQUITY UPLC BEH450 SEC 2.5 μm column (Waters).

To prepare the complex sample with phlorizin, the cells were solubilized for 1 h at 4 °C in buffer (50 mM HEPES–NaOH (pH 7.5), 300 mM NaCl, 2% (w/v) DDM (Calbiochem), protease inhibitor cocktail and 1 mM phlorizin). After ultracentrifugation (138,000$g$, 60 min, 4 °C), the supernatant was incubated with Affi-Gel 10 (Bio-Rad) coupled with a GFP-binding nanobody[37], and incubated for 2 h at 4 °C. The resin was washed five times with three column volumes of wash buffer (50 mM HEPES–NaOH (pH 7.5), 300 mM NaCl, 0.05% DDM (GLYCON Biochemicals) and 1 mM phlorizin), and gently suspended overnight with HRV3C protease to cleave the His10–sfGFP tag. After HRV3C protease digestion, the flow-through was pooled, concentrated, purified via size-exclusion chromatography on a Superose 6 Increase 10/300 GL column (GE Healthcare) and equilibrated with SEC buffer (20 mM HEPES–NaOH (pH 7.5), 150 mM NaCl, 0.03% DDM (GLYCON Biochemicals) and 0.5 mM phlorizin). For the samples complexed with canagliflozin, TA-1887, dapagliflozin and sotagliflozin, the same procedure was performed, but at concentrations of 30 μM of each inhibitor. The peak fractions were pooled and concentrated to 6–10 mg ml$^{-1}$.

### α-MG uptake in hSGLT2-transfected HEK293 cells

HEK293 cells (ECACC 85120602) were maintained in Dulbecco's modified Eagle medium (Gibco) supplemented with 10% fetal bovine serum (Thermo Scientific), 2 mM L-glutamine, 100 U ml$^{-1}$ benzylpenicillin and 100 μg ml$^{-1}$ streptomycin at 37 °C in a humidified atmosphere (5% $CO_2$ in air). HEK293 cells were seeded at $1.0 \times 10^5$ cells per well in poly-L-lysine coated 24-well plates. The cells in each well were transiently transfected with 0.25 μg hMAP17 plasmid and 0.50 μg hSGLT2 plasmid using Lipofectamine 2000 (Life Technologies) and cultured for 48 h. The medium was removed, and the cells were washed twice then preincubated with extracellular fluid buffer without glucose (122 mM NaCl, 25 mM NaHCO₃, 3 mM KCl, 1.4 mM CaCl₂, 2 mM MgSO₄, 0.4 mM K₂HPO₄ and 10 mM HEPES; pH 7.4) at 37 °C for 20 min. After preincubation, uptake was initiated by replacing the preincubation buffer with extracellular

fluid buffer containing 500 μM α-MG in the absence or presence of inhibitors. Uptake was completed by removing the uptake buffer and washing with ice-cold buffer three times, followed by solubilization in 1 N NaOH at room temperature. The increase in α-MG uptake was observed over 60 min (Supplementary Fig. 10), and the incubation time of the inhibition assay was 30 min.

Cell lysates were deproteinized by adding acetonitrile containing candesartan as the internal standard. The α-MG concentration was quantified via liquid chromatography–tandem mass spectrometry (LC–MS/MS) using the internal standard method.

Specific peaks of α-MG were observed in the lysates of mock and hSGLT2-expressing cells incubated with α-MG but not in those of mock cells in the absence of α-MG (Supplementary Fig. 10). Cellular protein content was determined using a bicinchoninic acid protein assay kit (Thermo Fisher Scientific). The uptake of α-MG was expressed as the ratio of concentration in the cells (in pmol per mg protein) to concentration in the medium (in pmol μl$^{-1}$); this is known as the cell-to-medium ratio (in μl per mg protein).

In the inhibition study, the cell-to-medium ratio of cells transfected with the empty vector was used as the background. The specific α-MG uptake was calculated by subtracting this background from the total cell-to-medium ratio and normalized to the uptake achieved without the inhibitor. $IC_{50}$ was calculated via nonlinear regression using GraphPad Prism 8.4.3.

### SGLT2 inhibitor-binding assay via affinity selection–mass spectrometry

To examine the inhibition of binding to the crude membrane, mammalian Expi293 cells were co-transfected with the hMAP17 and hSGLT2 plasmids, as described. The cells were collected and disrupted by sonication in a hypotonic buffer (50 mM HEPES–NaOH (pH 7.5), 10 mM KCl and protease inhibitor cocktail) or Na⁺-free hypotonic buffer (50 mM Tris–HCl (pH 7.5), 10 mM KCl and protease inhibitor cocktail). The cell debris were removed by centrifugation (2,000$g$, 5 min, 4 °C). The membrane fraction was collected by ultracentrifugation (112,000$g$, 30 min, 4 °C) and stored at −80 °C before use. The crude membrane (250 μg per sample) was incubated with SGLT2 inhibitor in an assay buffer (100 mM NaCl and 10 mM HEPES/Tris, pH 7.4) or Na⁺-free assay buffer (100 mM choline chloride and 10 mM HEPES/Tris, pH 7.4) at room temperature for 2 h. Reactions were terminated by filtration through a GF/C filter plate (Corning) presoaked in assay buffer containing 0.1% bovine serum albumin. The sample in the filter plate was washed three times with the assay buffer and eluted with acetonitrile: water (80:20, v/v). The extract solution from the filter plate was diluted with water containing candesartan as an internal standard, and the SGLT2 inhibitor concentration was quantified via LC–MS/MS.

Nonspecific binding was measured using the crude membrane of nontransfected Expi293 cells. Specific binding was calculated by subtracting nonspecific binding from the binding of hSGLT2-expressing cells. Specific binding was normalized to hSGLT2 protein expression levels, measured via FSEC. The equilibrium dissociation constant ($K_d$) and maximum number of binding sites ($B_{max}$) were calculated via nonlinear regression in GraphPad Prism 8.4.3. The specific binding of the hSGLT2 mutants was normalized to the $B_{max}$ of WT hSGLT2.

### Quantification of SGLT2 substrate and inhibitors via LC–MS/MS

The concentrations of the extract solution from the filter plate and of the cell lysate were quantified using a tandem mass spectrometry QTRAP6500 System (SCIEX) coupled with an ACQUITY UPLC system (Waters) using the internal standard method. Mobile phases A and B used 10 mM of ammonium bicarbonate and acetonitrile, respectively. Chromatographic separation was performed on an ACQUITY UPLC BEH C18 column (2.1 mm × 100 mm, 1.7 μm; Waters) at 50 °C, with the following gradient of mobile phase B: 1% (at 0.00 to 0.50 min), 1% to 95% (0.50 to 2.00 min), 95% (2.00 to 2.50 min) and 1% (2.51 to 3.00 min);

the flow rate was 0.4 ml min$^{-1}$. Mass spectrometric detection was performed by multiple reaction monitoring in the electrospray-ionization negative-ion mode controlled by Analyst 1.6.2, using $m/z$ 443.1/364.9 for canagliflozin; 425.9/264.1 for TA-1887; 407.0/328.8 for dapagliflozin; 423.0/387.0 for sotagliflozin; 435.0/273.0 for phlorizin; 273.0/148.9 for phloretin; 192.9/100.9 for α-MG; and 439.0/309.1 for candesartan.

### Electron microscopy sample preparation

The purified protein solution of hSGLT2–MAP17 was mixed with the inhibitor solutions (except for phlorizin), at final concentrations of 0.5 mM dapagliflozin, TA-1887, sotagliflozin or canagliflozin. After incubation for 1 h on ice, the grids were glow-discharged in low-pressure air at a 10 mA current in a PIB-10 (Vacuum Device). The protein solutions containing 0.5 mM of the inhibitors were applied to a freshly glow-discharged Quantifoil Holey Carbon Grid (R1.2/1.3, Cu/Rh, 300 mesh) (SPT Labtech) using a Vitrobot Mark IV system (Thermo Fisher Scientific) at 4 °C, with a blotting time of 4–6 s under 99% humidity; the grids were then plunge-frozen in liquid ethane.

### Electron microscopy data collection and processing

The grids containing phlorizin, TA-1887, dapagliflozin and sotagliflozin were transferred to a Titan Krios G3i system (Thermo Fisher Scientific) running at 300 kV and equipped with a Gatan Quantum-LS Energy Filter (GIF) and a Gatan K3 Summit direct electron-detector in correlated double-sampling mode. Imaging was performed at a nominal magnification of 105,000×, corresponding to a calibrated pixel size of 0.83 Å per pixel, at the University of Tokyo, Japan. Each movie was dose-fractionated to 64 frames at a dose rate of 6.2–9.0 e$^-$ per pixel per second at the detector, resulting in a total accumulated exposure of 64 e$^-$ Å$^{-2}$ of the specimen. The data were automatically acquired using the image-shift method in SerialEM software[38], with a defocus range of −0.8 to −1.6 μm.

The grid with canagliflozin was transferred to a Titan Krios G4 device (Thermo Fisher Scientific) running at 300 kV and equipped with a Gatan Quantum-LS Energy Filter (GIF) and a Gatan K3 Summit direct electron-detector in correlated double-sampling mode. Imaging was performed at a nominal magnification of 215,000×, corresponding to a calibrated pixel size of 0.4 Å per pixel, at the University of Tokyo, Japan. Each movie was recorded for 1.4 s and subdivided into 64 frames. Electron flux was set to 7.5 e$^-$ per pixel per second at the detector, resulting in an accumulated exposure of 64 e$^-$ Å$^{-2}$ of the specimen. The data were automatically acquired via the image-shift method using EPU software (Thermo Fisher Scientific), with a defocus range of −0.6 to −1.6 μm. The total number of images is described in Table 1.

For all datasets, the dose-fractionated movies were subjected to beam-induced motion correction using RELION[24], and the contrast transfer function (CTF) parameters were estimated using CTFFIND4 (ref. 39).

For the canagliflozin-bound state dataset, 2,364,108 particles were initially selected from 19,943 micrographs using the topaz-picking function in RELION-4.0 (ref. 25). Particles were extracted by downsampling to a pixel size of 3.2 Å per pixel. These particles were subjected to several rounds of 2D and 3D classification. The best class contained 221,701 particles, which were then re-extracted with a pixel size of 1.60 Å per pixel and subjected to 3D refinement. The second 3D classification resulted in three map classes. The best class from the 3D classification contained 179,761 particles, which were subjected to 3D refinement. The particles were subsequently subjected to micelle subtraction and non-aligned 3D classification using a mask (without micelles), resulting in three map classes. The best class, containing 65,919 particles, was subjected to 3D refinement, reversion to the original particles, and 3D refinement. The particle set was resized to 1.00 Å per pixel and subjected to Bayesian polishing, 3D refinement and per-particle CTF refinement before the final 3D refinement and post-processing, yielding a map with a global resolution of 3.1 Å, according to the FSC 0.143 criterion. Finally, local

resolution was estimated using RELION-4. The processing strategy is illustrated in Supplementary Fig. 2.

For the dapagliflozin-bound-state dataset, 3,692,950 particles were initially selected from 4,841 micrographs using the topaz-picking function in RELION-4.0. Particles were extracted by downsampling to a pixel size of 3.32 Å per pixel. These particles were subjected to several rounds of 2D and 3D classification. The best class contained 569,516 particles, which were then re-extracted at a pixel size of 1.30 Å per pixel and subjected to 3D refinement. Non-aligned 3D classification using a soft mask covering the proteins and micelles resulted in four map classes. The best class from the 3D classification contained 197,695 particles, which were subjected to 3D refinement, per-particle CTF refinement, and 3D refinement. The resulting 3D model and particle set were resized to 1.11 Å per pixel and subjected to Bayesian polishing, 3D refinement and per-particle CTF refinement. Final 3D refinement and post-processing yielded maps with global resolutions of 2.8 Å, according to the FSC 0.143 criterion. Finally, the local resolution was estimated using RELION-3. The processing strategy is illustrated in Supplementary Fig. 3.

For the TA-1887-bound state dataset, 3,395,470 particles were initially selected from 4,383 micrographs using the topaz-picking function in RELION-4. Particles were extracted by downsampling to a pixel size of 3.32 Å per pixel. These particles were subjected to several rounds of 2D and 3D classification. The best class contained 274,477 particles, which were then re-extracted with a pixel size of 1.30 Å per pixel and subjected to 3D refinement. Non-aligned 3D classification using a soft mask covering the proteins and micelles resulted in three map classes. The best class from the 3D classification contained 103,853 particles, which were subjected to 3D refinement, per-particle CTF refinement, and 3D refinement. The resulting 3D model and particle set were resized to 1.11 Å per pixel and subjected to Bayesian polishing, 3D refinement and per-particle CTF refinement. Final 3D refinement and post-processing yielded maps with global resolutions of 2.9 Å, according to the FSC 0.143 criterion. Local resolution was estimated using RELION-4. The processing strategy is illustrated in Supplementary Fig. 4.

For the sotagliflozin-bound state dataset, 5,242,427 particles were initially selected from 5,499 micrographs using the topaz-picking function in RELION-4. Particles were extracted by downsampling to a pixel size of 3.32 Å per pixel. These particles were subjected to several rounds of 2D and 3D classifications. The best class contained 823,369 particles, which were then re-extracted with a pixel size of 1.30 Å per pixel and subjected to 3D refinement. Non-aligned 3D classification using a soft mask covering the proteins and micelles resulted in four classes of maps. The two good classes from the 3D classification contained 227,811 particles, which were subjected to 3D refinement. The resulting 3D model and particle set were resized to 1.11 Å/px and subjected to Bayesian polishing, 3D refinement and further non-aligned 3D classification using a soft mask covering the proteins and micelles. The best class from the 3D classification contained 72,773 particles, which were subjected to 3D refinement, per-particle CTF refinement, 3D refinement, Bayesian polishing, 3D refinement and per-particle CTF refinement. The final 3D refinement and post-processing yielded maps with global resolutions of 3.1 Å, according to the FSC 0.143 criterion. Finally, the local resolution was estimated using RELION. The processing strategy is illustrated in Supplementary Fig. 5.

For the phlorizin-bound state dataset, 3,013,029 particles were initially selected from 3,159 micrographs using the Laplacian-of-Gaussian picking function in RELION-3.1 (ref. 24) and were used to generate 2D models for reference-based particle picking. Particles were extracted by downsampling to a pixel size of 3.32 Å per pixel. These particles were subjected to several rounds of 2D and 3D classification. The best class contained 324,355 particles, which were then re-extracted with a pixel size of 1.66 Å per pixel and subjected to 3D refinement. The particles were subsequently subjected to micelle subtraction and non-aligned 3D classification using a mask (excluding the micelles), resulting in three

map classes. The best class contained 76,485 particles, which were then subjected to 3D refinement and reversion to the original particles. The particle set was resized to 1.30 Å per pixel, and subjected to Bayesian polishing, 3D refinement and per-particle CTF refinement before the final 3D refinement and post-processing, yielding a map with a global resolution of 3.3 Å, according to the FSC 0.143 criterion. Finally, the local resolution was estimated using RELION-3. The processing strategy is illustrated in Supplementary Fig. 6.

### Model building and validation
The models of the phlorizin-bound inward state of hSGLT2–MAP17 were manually built, de novo, using the cryo-EM density map tool in COOT[40], facilitated by an hSGLT2-homology model generated using Alphafold2 (ref. [41]). After manual adjustment, the models were subjected to structural refinement via the Servalcat pipeline in REFMAC5 (ref. [42]) and manual real-space refinement in COOT. The models of the dapagliflozin-, TA-1887-, sotagliflozin- and canagliflozin-bound outward states were built using the Alphafold2-derived hSGLT2-homology model as the starting model. The 3D reconstruction and model refinement statistics are summarized in Table 1. All molecular graphics figures were prepared using CueMol (http://www.cuemol.org) and UCSF Chimera[43].

### Thermostability measurement
The thermostability of the detergent-solubilized proteins was analyzed using an FSEC-based thermostability assay[44]. Mammalian Expi293 cells (Thermo Fisher Scientific) were transiently transfected with the plasmids and with ExpiFectamine (Thermo Fisher Scientific). For each 1 ml transfection, 1 ml of cells ($2.4 \times 10^6$ cells) was transferred to each well in a 96-well MASTERBLOCK (Greiner Bio-One). For the hSGLT2–MAP17 heterodimer, 0.8 μg hSGLT2 plasmid and 0.4 μg MAP17 plasmid were mixed in Opti-MEM (total volume 60 μl) and 3.2 μl ExpiFectamine 293 Reagent was added to 56.8 μl Opti-MEM. For hSGLT2 alone, 1.2 μg hSGLT2 plasmid was added to Opti-MEM (total volume 60 μl). After incubating for 5 min at room temperature, the diluted plasmid was added to the diluted ExpiFectamine 293 Reagent, gently mixed, and incubated for 20 min at room temperature. The reagent–plasmid mixture was added to each well and incubated at 37 °C in the presence of 8% $CO_2$ on a Maximizer MBR-022UP bioshaker (TAITEC) at 1,200 rpm. After 48 h incubation at 37 °C, the cells were collected from 6 ml of the culture via centrifugation. The cell pellet was resuspended in 600 μl of buffer (50 mM HEPES–NaOH (pH 7.5), 300 mM NaCl, 1% (w/v) DDM and protease inhibitor cocktail) and shaken for 60 min at 4 °C. After clearing the cell lysate via centrifugation (20,000g, 30 min, 4 °C), 45 μl portions of the cell lysate were incubated at 4, 10, 15, 20, 25, 30, 35, 40, 45, 50, 55 or 60 °C for 10 min in a PCR Thermal Cycler SP (Takara Bio). The sample was again centrifuged (20,000g, 60 min, 4 °C) to clear the lysate, and a 1 μl portion of the supernatant was placed in an ACQUITY UPLC BEH450 SEC 2.5 μm column (Waters), pre-equilibrated with buffer containing 50 mM Tris, pH 7.6, 150 mM NaCl and 0.05% DDM (GLYCON Biochemicals). The GFP fluorescence of the eluent was monitored, and the peak heights of heat-treated samples were normalized to that of the untreated sample. For each mutant, the measurement was performed at least three times, and the melting temperatures were determined by fitting the curves to a sigmoidal dose–response equation, using GraphPad Prism 7.

### Water analysis using 3D-RISM
The structure was preprocessed using the Protein Preparation Wizard[45] in Schrödinger Suite v2021-4 (Schrödinger). The default operation flipped the carbamoyl group of Q457 in the dapagliflozin complex; this was therefore corrected manually. All water in the pocket was retained. Finally, restrained minimization was performed using the OPLS4 force field[46]. Water analysis was performed on the prepared structures using 3D-RISM (as implemented in MOE (Molecular Operating Environment)). The supplementary program required for water site analysis is available directly from Chemical Computing Group ULC.

### Reporting summary
Further information on research design is available in the Nature Portfolio Reporting Summary linked to this article.

## Data availability
The cryo-EM density maps were deposited in the Electron Microscopy Data Bank under accession codes EMD-34673 (canagliflozin-bound state), EMD-34705 (dapagliflozin-bound state), EMD-34610 (TA-1887-bound state), EMD-34737 (sotagliflozin-bound state) and EMD-34823 (phlorizin-bound state). The atomic coordinates have been deposited in the Protein Data Bank under IDs 8HDH (canagliflozin-bound state), 8HEZ (dapagliflozin-bound state), 8HB0 (TA-1887-bound state), 8HG7 (sotagliflozin-bound state) and 8HIN (phlorizin-bound state). Source data are provided with this paper.

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

## Acknowledgements

We thank Y. Lee and T. Nishizawa for technical advice on sample preparation; K. Yamashita for technical advice on refinement using Servalcat; T. Saijo, T. Takahashi and Y. Kamikozawa for technical advice on functional analyses; and H. Nishimasu, M. Shiotani, C. Kuriyama and Y. Yamamoto for fruitful discussions about the inhibitory mechanism and paper preparation. We thank the scientific staff of the cryo-EM facility of the University of Tokyo, and especially Y. Kise, Y. Sakamaki, T. Kusakizako, H. Yanagisawa, A. Tsutsumi, M. Kikkawa and R. Danev. The authors received no specific funding for this work.

## Author contributions

M.H. designed the entire study. M.H. performed the cryo-EM analyses, with sample preparation assistance from T.K. M.H. and T.A. designed

and performed the functional analyses, with sample preparation assistance from M.H. and T.K. M.H. performed model building and model refinement, with assistance from H.K. and K.M. M.H., T.A., N.T., I.M. and O.N. wrote and edited the manuscript, with help from all other authors. M.H., T.A., I.M. and O.N. supervised the study.

## Competing interests

M.H., T.A., K.M., H.K., T.K., N.T. and I.M. are employees of Mitsubishi Tanabe Pharma Corporation. O.N. is a co-founder of, and scientific advisor to, Curreio, Tokyo, Japan.

## Additional information

**Extended data** is available for this paper at https://doi.org/10.1038/s41594-023-01134-0.

**Correspondence and requests for materials** should be addressed to Masahiro Hiraizumi, Osamu Nureki or Ikuko Miyaguchi.

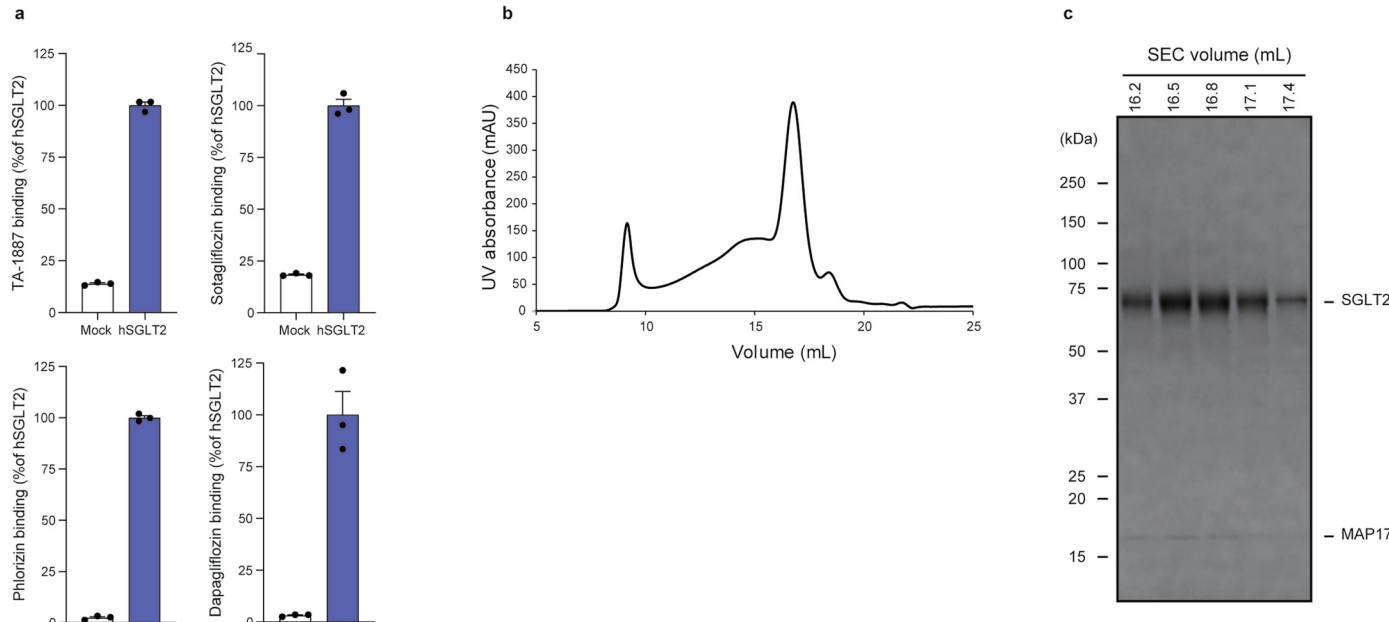

**Extended Data Fig. 1 | Biochemical characterization of the hSGLT2–MAP17 complex.** a, SGLT2 inhibitors (10 nM TA-1887, 50 nM sotagliflozin, 100 nM phlorizin and 10 nM dapagliflozin) binding to the crude membrane expressing hSGLT2 and MAP17. Data are shown as mean ± SEM (n = 3, technical replicates).

b, Representative size-exclusion chromatography profile of hSGLT2–MAP17. c, SDS-PAGE analysis of the hSGLT2–MAP17 peak fractions via size-exclusion chromatography (SEC) purification.

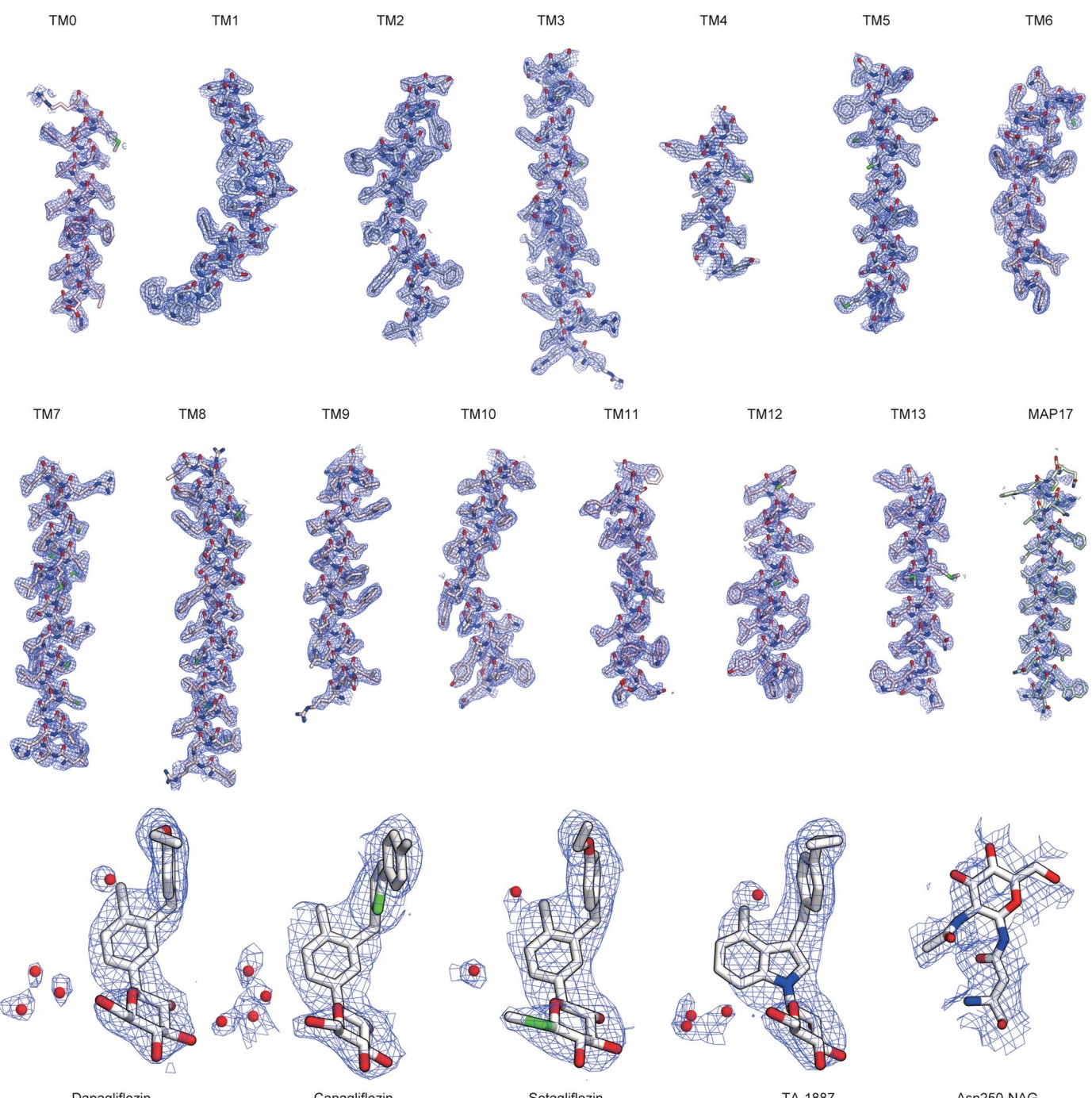

**Extended Data Fig. 2 | The outward-facing model of hSGLT2–MAP17 in the density maps.** The cryo-EM density and atomic model of each segment of the outward-facing model hSGLT2–MAP17, inhibitors, and glycosylation sites, contoured to 3.0 σ, 4.0 σ, and 2.0 σ, respectively. Red spheres: water molecules around the inhibitors.

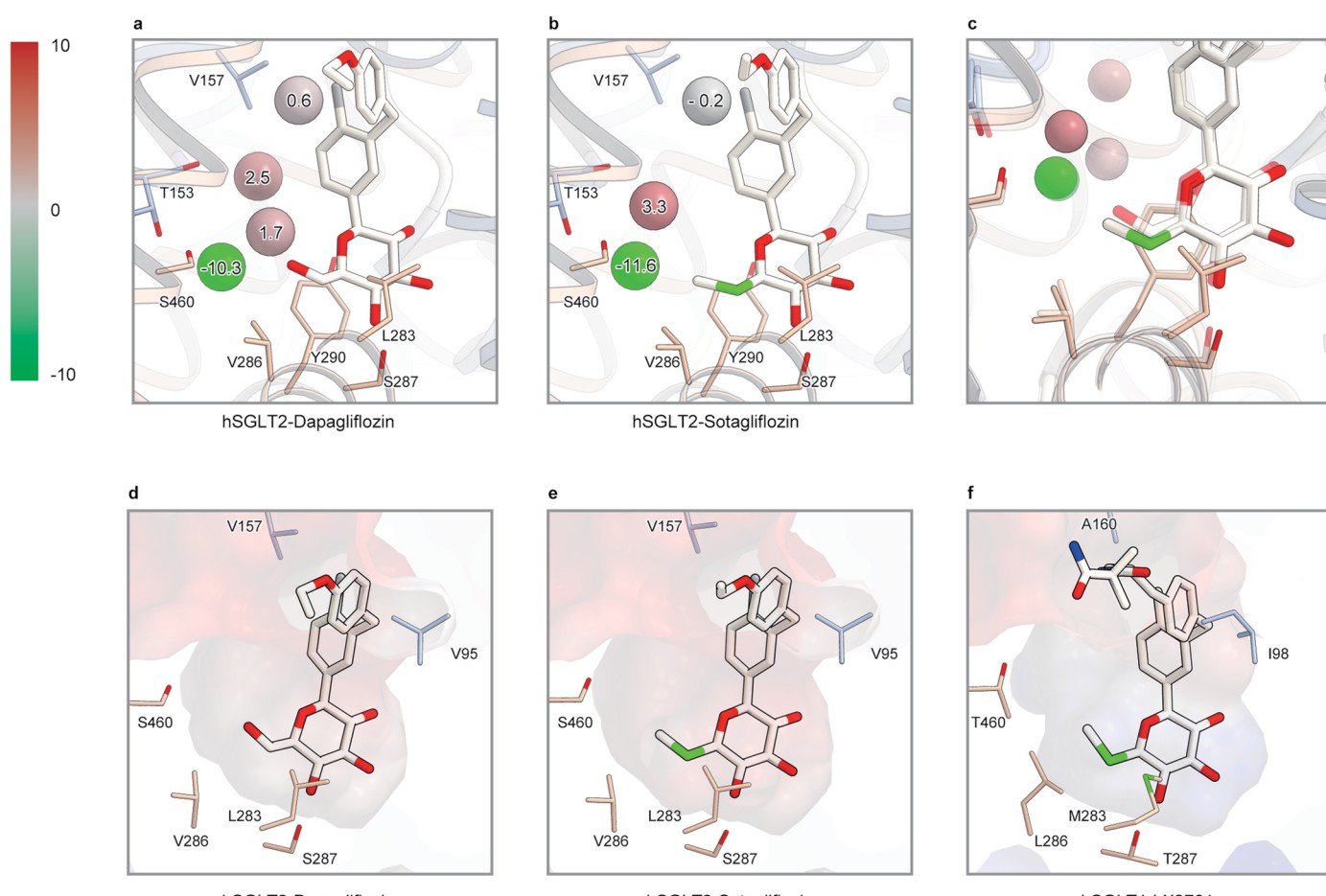

**Extended Data Fig. 3 | Analysis of gliflozin binding sites in SGLT1/2.** 3D-RISM analysis of hSGLT2-dapagliflozin (a) and hSGLT-sotagliflozin (b) complexes. Energetically unstable water molecules predicted by 3D-RISM are red, and energetically stable water molecules are green. c, Overlay of hSGLT2-dapagliflozin and hSGLT-sotagliflozin complexes with water molecules. d-f, Structural comparison of gliflozins binding sites in hSGLT1/2. Cross-sections of the electrostatic surface potentials surrounding gliflozins are displayed as a color gradient from red (negative) to blue (positive). Residues that differ between SLGT1 and SGLT2 in the binding site are shown in ballstick.

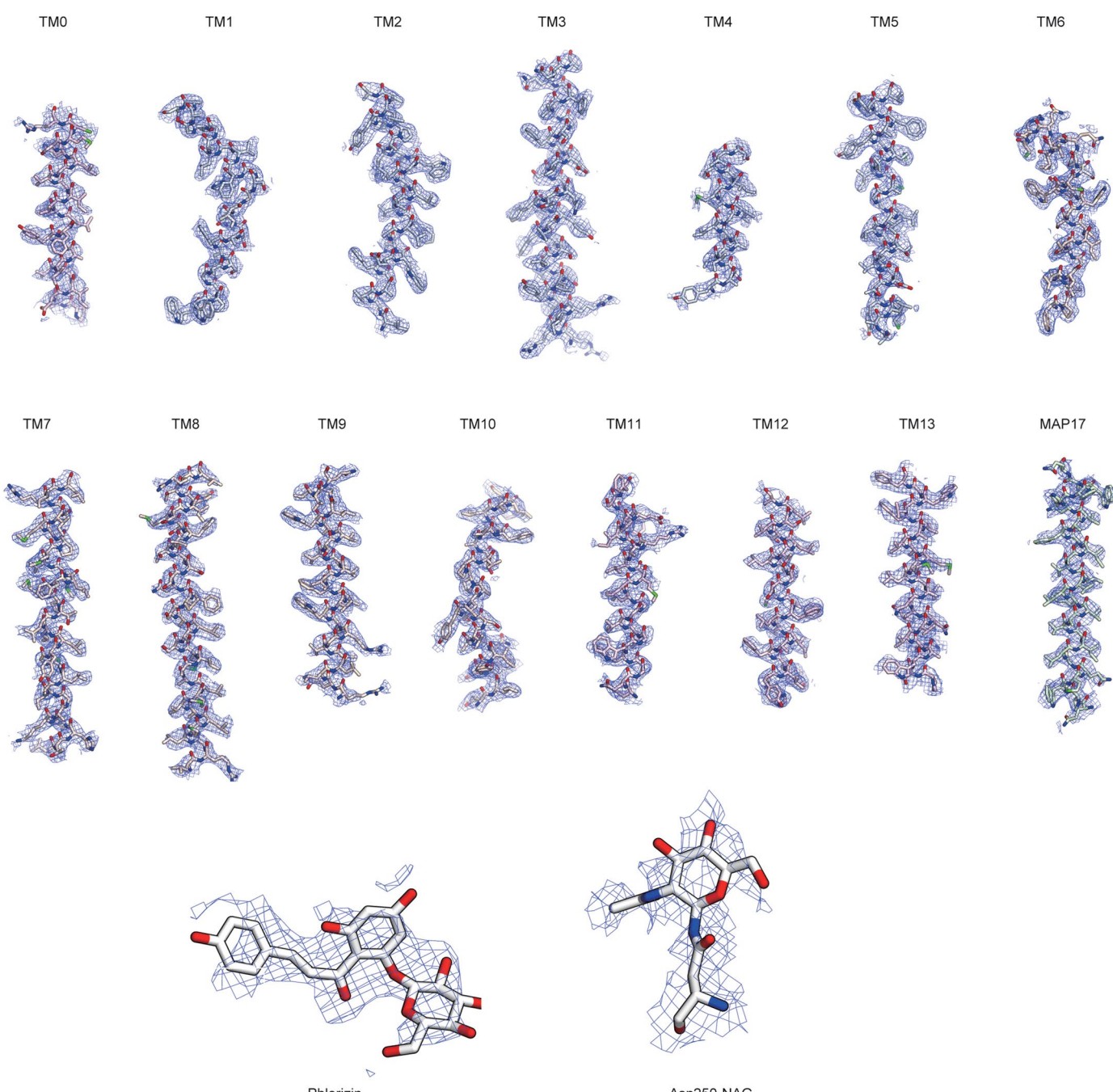

**Extended Data Fig. 4 | Inward-open model of hSGLT2–MAP17 in the density maps.** The cryo-EM density and atomic models of each segment of the inward-open model of hSGLT2–MAP17, phlorizin, and glycosylation sites, contoured to 2.9 σ, 2.7 σ, and 2.7 σ, respectively.

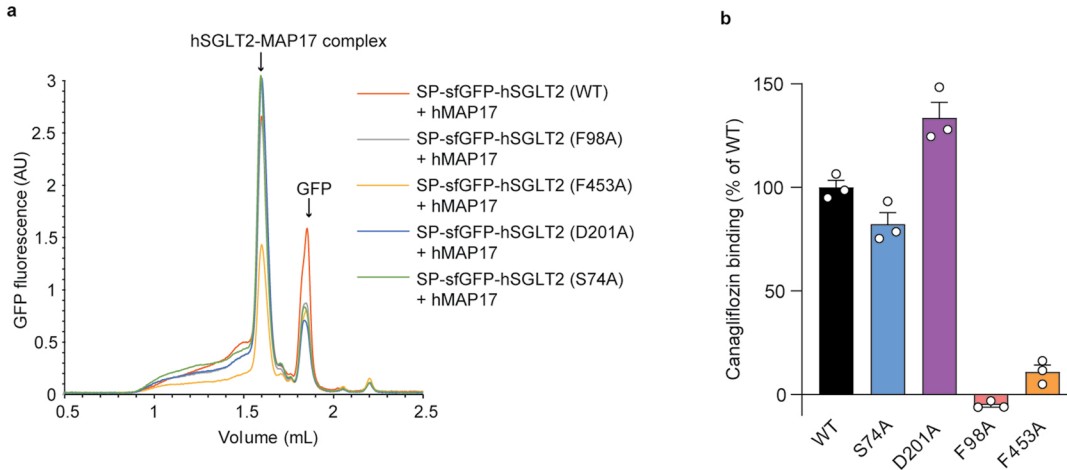

**Extended Data Fig. 5 | Profiles for various mutations of sfGFP-tagged hSGLT2 with MAP17.** a, FSEC profiles for various mutations of sfGFP-tagged hSGLT2 with MAP17. The arrows indicate the elution positions of the hSGLT2–MAP17 heterodimer and free GFP. b, Canagliflozin binding to the crude membrane expressing wild-type hSGLT2 and mutants. Crude membranes were incubated with 30 nM canagliflozin. Binding was measured via LC–MS/MS. Data are shown as mean ± SEM (n = 3, technical replicates).

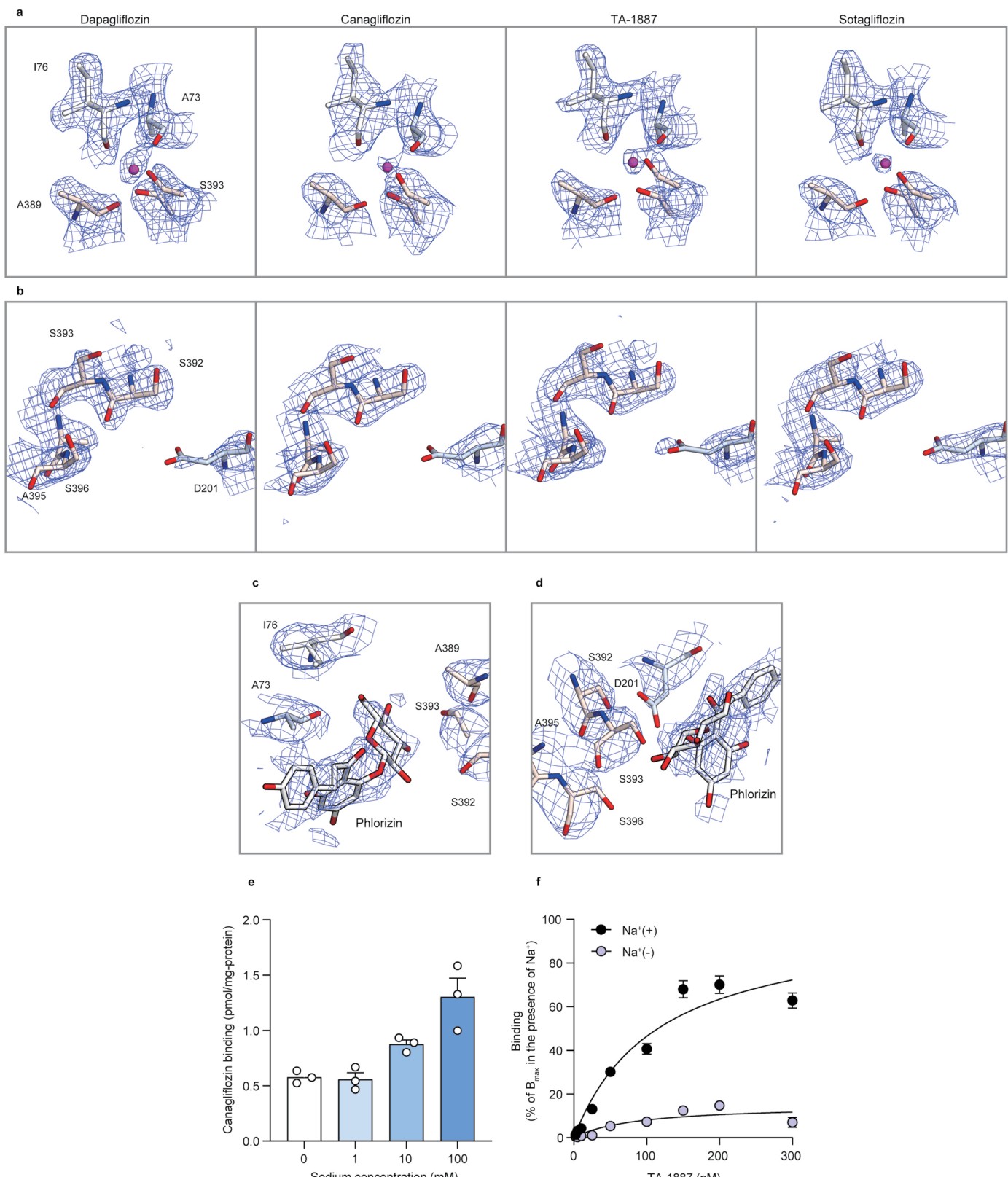

**Extended Data Fig. 6 | Sodium ion-binding sites of SGLT2 and sodium-dependent binding of SGLT2 inhibitors.** a, The sodium ion-binding Na2 sites of dapagliflozin, canagliflozin, TA-1887, and sotagliflozin are shown. b, Sites in SGLT2 corresponding to Na3 sites where the other sodium ion binds in SGLT1. No electron density corresponding to a sodium ion can be observed. c–d, The sodium ion-binding Na2 sites, and the sites corresponding to Na3 sites of the phlorizin-bound inward-open conformation. The phlorizin-binding site is near the Na2 and Na3 sites. e, Sodium concentration-dependent binding of canagliflozin. Each point represents the mean ± SEM (n = 3, technical replicates). f, Concentration-dependent binding of TA-1887 in the presence and absence of Na+. Each point represents the mean ± SEM (n = 3, technical replicates).

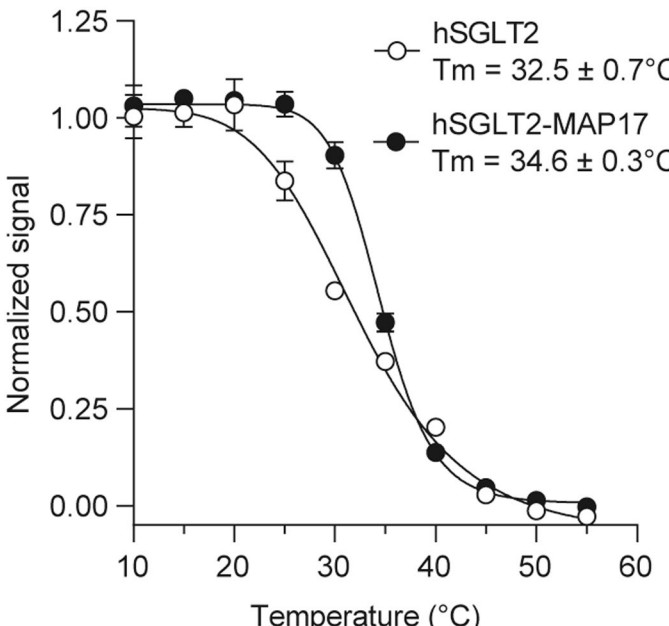

**Extended Data Fig. 7 | FSEC-based thermal shift assay using sfGFP-tagged hSGLT2 in the presence or absence of MAP17.** Each point represents the mean ± SEM (n = 3, technical replicates).

**Extended Data Table 1 | Kinetic parameters of phlorizin and phloretin binding to wild-type or mutated hSGLT2**

| | Phlorizin | | | |
|---|---|---|---|---|
| Construct | $K_{d1}$ (nM) | $B_{max1}$ (pmol/mg-protein) | $K_{d2}$ (nM) | $B_{max2}$ (pmol/mg-protein) |
| WT | 28.1 ± 11.3 | 0.356 ± 0.069 | 12400 ± 6900 | 16.8 ± 7.4 |
| S74A | 113 ± 16 | 1.16 ± 0.07 | ND | ND |
| D201A | 72.2 ± 10.0 | 0.550 ± 0.028 | ND | ND |
| F98A | ND | ND | ND | ND |
| F453A | ND | ND | ND | ND |

| | Phloretin | |
|---|---|---|
| Construct | $K_d$ (nM) | $B_{max}$ (pmol/mg-protein) |
| WT | 12000 ± 1900 | 51.7 ± 3.46 |
| S74A | ND | ND |
| D201A | ND | ND |

Data were represented in mean±SEM (n=3 (phlorizin) and 4 (phloretin), technical replicates). ND; not determined.

**Extended Data Table 2 | IC50 values of phlorizin in α-MG uptake by wild-type and mutant hSGLT2**

| Construct | logIC50 | IC50 (nM) |
|-----------|-----------|-----------|
| WT | 1.75 ± 0.07 | 57.0 |
| S74A | 2.16 ± 0.08 | 145 |
| F98A | 3.22 ± 0.47 | 1640 |
| F453A | 2.67 ± 0.19 | 268 |

Data were represented in mean±SEM (n=3, biological replicates).

**Extended Data Table 3 | Kinetic parameters of canagliflozin and TA-1887 binding in the membrane fraction of hSGLT2 and MAP17 expressing cells, in the presence or absence of Na⁺**

|  |  | Na(+) | Na(-) |
|---|---|---|---|
| Canagliflozin | $B_{max}$ (pmol/mg-protein) | 1.91 ± 0.16 | 1.33 ± 0.12 |
|  | $K_d$ (nM) | 94.0 ± 21.8 | 109 ± 25.4 |
| TA-1887 | $B_{max}$ (pmol/mg-protein) | 4.67 ± 0.49 | 0.705 ± 0.155 |
|  | $K_d$ (nM) | 115 ± 28 | 86.4 ± 50.0 |

Data were represented in mean±SEM (n=3, technical replicates).

Masahiro Hiraizumi
Osamu Nureki

# Reporting Summary

## Statistics

For all statistical analyses, confirm that the following items are present in the figure legend, table legend, main text, or Methods section.

| n/a | Confirmed | |
|---|---|---|
| ☐ | ☒ | The exact sample size ($n$) for each experimental group/condition, given as a discrete number and unit of measurement |
| ☐ | ☒ | A statement on whether measurements were taken from distinct samples or whether the same sample was measured repeatedly |
| ☒ | ☐ | The statistical test(s) used AND whether they are one- or two-sided<br>*Only common tests should be described solely by name; describe more complex techniques in the Methods section.* |
| ☒ | ☐ | A description of all covariates tested |
| ☒ | ☐ | A description of any assumptions or corrections, such as tests of normality and adjustment for multiple comparisons |
| ☐ | ☒ | A full description of the statistical parameters including central tendency (e.g. means) or other basic estimates (e.g. regression coefficient) AND variation (e.g. standard deviation) or associated estimates of uncertainty (e.g. confidence intervals) |
| ☒ | ☐ | For null hypothesis testing, the test statistic (e.g. $F$, $t$, $r$) with confidence intervals, effect sizes, degrees of freedom and $P$ value noted<br>*Give P values as exact values whenever suitable.* |
| ☒ | ☐ | For Bayesian analysis, information on the choice of priors and Markov chain Monte Carlo settings |
| ☒ | ☐ | For hierarchical and complex designs, identification of the appropriate level for tests and full reporting of outcomes |
| ☒ | ☐ | Estimates of effect sizes (e.g. Cohen's $d$, Pearson's $r$), indicating how they were calculated |

*Our web collection on statistics for biologists contains articles on many of the points above.*

## Software and code

Policy information about availability of computer code

| Data collection | EPU (version 1.19), Serial EM (version 3.7.10), Analyst 1.6.2. |
|---|---|
| Data analysis | RELION (version 3.1 and version 4.0), Servalcat (version 0.2.0), MOLREP (version 11.7), COOT (version 0.8.9.1), UCSF Chimera (version 1.14), CueMol2 (http://www.cuemol.org/ version 2.2.3.443), GraphPad Prism 8.4.3., REFMAC5(version 5.8.0267) |

For manuscripts utilizing custom algorithms or software that are central to the research but not yet described in published literature, software must be made available to editors and reviewers. We strongly encourage code deposition in a community repository (e.g. GitHub). See the Nature Portfolio guidelines for submitting code & software for further information.

## Data

Policy information about availability of data

All manuscripts must include a data availability statement. This statement should provide the following information, where applicable:
- Accession codes, unique identifiers, or web links for publicly available datasets
- A description of any restrictions on data availability
- For clinical datasets or third party data, please ensure that the statement adheres to our policy

Cryo-EM density maps were deposited in the Electron Microscopy Data Bank under accession codes EMD-34673 (canagliflozin-bound state), EMD-34705 (dapagliflozin-bound state), EMD-34610 (TA-1887-bound state), EMD-34737 (sotagliflozin-bound state), and EMD-34823 (phlorizin-bound state). Atomic

coordinates have been deposited in the Protein Data Bank under IDs 8HDH (canagliflozin-bound state), 8HEZ (dapagliflozin-bound state), 8HB0 (TA-1887-bound state), 8HG7 (sotagliflozin-bound state), and 8HIN (phlorizin-bound state). Source data are provided with the paper.

# Research involving human participants, their data, or biological material

Policy information about studies with human participants or human data. See also policy information about sex, gender (identity/presentation), and sexual orientation and race, ethnicity and racism.

| Reporting on sex and gender | not relevant |
|---|---|
| Reporting on race, ethnicity, or other socially relevant groupings | not relevant |
| Population characteristics | not relevant |
| Recruitment | not relevant |
| Ethics oversight | not relevant |

Note that full information on the approval of the study protocol must also be provided in the manuscript.

# Field-specific reporting

Please select the one below that is the best fit for your research. If you are not sure, read the appropriate sections before making your selection.

☒ Life sciences          ☐ Behavioural & social sciences          ☐ Ecological, evolutionary & environmental sciences

For a reference copy of the document with all sections, see nature.com/documents/nr-reporting-summary-flat.pdf

# Life sciences study design

All studies must disclose on these points even when the disclosure is negative.

| Sample size | No statistical method was used to determine the sample size. For cryo-EM analyses, sample sizes were determined by the availability of microscope time and the number of particles on electron microscopy grids enough to obtain a structure at the reported resolution. For transporter functional analyses, no sample size determination analysis was determined as binding assay and uptake assay got consistent signal and low background to ensure the reproducibility and no statistical analysis was performed. These assays are performed with similar sample sizes in previous studies as well. |
|---|---|
| Data exclusions | For cryo-EM analyses, particles that did not contribute to improving map quality were excluded following the standard classification procedures in RELION. This is standard practice for structure determination by cryo-EM. |
| Replication | For cryo-EM analyses, related experiments including FSEC, purification, and SDS-PAGE were reproduced at least two times and structure determination was completed once. In transporter functional analyses, three or four technical replicates of binding assay and three or four biologically independent replicates of uptake assay was evaluated and all attempts at replication were successful. |
| Randomization | For cryo-EM analyses, particles were randomly assigned to half-maps for resolution determination following the standard procedures in RELION. For transporter functional analyses, randomization was not performed since samples were not divided into two or more groups. |
| Blinding | For cryo-EM analyses and transporter functional analyses, blinding was not applicable since this type of studies does not use group allocation. |

# Reporting for specific materials, systems and methods

We require information from authors about some types of materials, experimental systems and methods used in many studies. Here, indicate whether each material, system or method listed is relevant to your study. If you are not sure if a list item applies to your research, read the appropriate section before selecting a response.

## Materials & experimental systems

| n/a | Involved in the study |
|---|---|
| ☒ | ☐ Antibodies |
| ☐ | ☒ Eukaryotic cell lines |
| ☒ | ☐ Palaeontology and archaeology |
| ☒ | ☐ Animals and other organisms |
| ☒ | ☐ Clinical data |
| ☒ | ☐ Dual use research of concern |
| ☒ | ☐ Plants |

## Methods

| n/a | Involved in the study |
|---|---|
| ☒ | ☐ ChIP-seq |
| ☒ | ☐ Flow cytometry |
| ☒ | ☐ MRI-based neuroimaging |

## Eukaryotic cell lines

Policy information about cell lines and Sex and Gender in Research

| | |
|---|---|
| Cell line source(s) | HEK293 (ECACC) and Expi293 (Thermo Fisher Scientific) |
| Authentication | No further authentication was performed after receipt. |
| Mycoplasma contamination | Not performed |
| Commonly misidentified lines (See ICLAC register) | No commonly misidenfied cell lines were used. |

