## [Peer Review File · Nature Structural & Molecular Biology]

Peer Review Information

Manuscript Title: Structural insights into the mechanism of the human SGLT2–MAP17 glucose transporter

Corresponding author name(s): Ikuko Miyaguchi, Masahiro Hiraizumi

Reviewer Comments & Decisions:

Decision Letter, initial version:

Message: 2nd Mar 2023

Dear Dr. Miyaguchi,

Thank you again for submitting your manuscript "Structural insights into the mechanism of the human SGLT2–MAP17 glucose transporter". I apologize for the delay in responding, which resulted from the difficulty in obtaining suitable referee reports. Nevertheless, we now have comments (below) from the 3 reviewers who evaluated your paper. In light of those reports, we remain interested in your study and would like to see your response to the comments of the referees, in the form of a revised manuscript.

You will see that while all reviewers appreciate the presented results, they raise multiple concerns which we would like to see addressed in a revision. Specifically, reviewers suggest expanding discussion of several points - comparison of reported structures to SGLT1, further explanation of Na⁺ stabilising mechanism and the unexpected binding of phlorizin in the inward facing conformation, for example. Additionally, reviewer #3 requests that enhancement of SGLT2 activity by MAP17 is further probed mechanistically. In our editorial view, we agree that this would add to the impact and value of the manuscript, however we would not insist on this being addressed experimentally.

Please be sure to address/respond to all concerns of the referees in full in a point-by-point response and highlight all changes in the revised manuscript text file. If you have comments that are intended for editors only, please include those in a separate cover letter.

We expect to see your revised manuscript within 6 weeks. If you cannot send it within this time, please contact us to discuss an extension; we would still consider your revision, provided that no similar work has been accepted for publication at NSMB or published elsewhere.

Reporting Summary:

When submitting the revised version of your manuscript, please pay close attention to our [href="https://www.nature.com/nature-portfolio/editorial-policies/image-integrity">Digital Image Integrity Guidelines. and to the following points below:](https://www.nature.com/nature-portfolio/editorial-policies/image-integrity)

Please note that all key data shown in the main figures as cropped gels or blots should be presented in uncropped form, with molecular weight markers. These data can be aggregated into a single supplementary figure item. While these data can be displayed in a relatively informal style, they must refer back to the relevant figures. These data should be submitted with the final revision, as source data, prior to acceptance, but you may want to start putting it together at this point.

SOURCE DATA: we urge authors to provide, in tabular form, the data underlying the graphical representations used in figures. This is to further increase transparency in data reporting, as detailed in this editorial (<http://www.nature.com/nsmb/journal/v22/n10/full/nsmb.3110.html>). Spreadsheets can be submitted in excel format. Only one (1) file per figure is permitted; thus, for multi-

paneled figures, the source data for each panel should be clearly labeled in the Excel file; alternately the data can be provided as multiple, clearly labeled sheets in an Excel file. When submitting files, the title field should indicate which figure the source data pertains to. We encourage our authors to provide source data at the revision stage, so that they are part of the peer-review process.

Data availability: this journal strongly supports public availability of data. All data used in accepted papers should be available via a public data repository, or alternatively, as Supplementary Information. If data can only be shared on request, please explain why in your Data Availability Statement, and also in the correspondence with your editor. Please note that for some data types, deposition in a public repository is mandatory - more information on our data deposition policies and available repositories can be found below: <https://www.nature.com/nature-research/editorial-policies/reporting-standards#availability-of-data>

Nature Structural & Molecular Biology is committed to improving transparency in authorship. As part of our efforts in this direction, we are now requesting that all authors identified as 'corresponding author' on published papers create and link their Open Researcher and Contributor Identifier (ORCID) with their account on the Manuscript Tracking System (MTS), prior to acceptance. This applies to primary research papers only. ORCID helps the scientific community achieve unambiguous attribution of all scholarly contributions. You can create and link your ORCID from the home page of the MTS by clicking on 'Modify my Springer Nature account'. For more information please visit please visit www.springernature.com/orcid.

[redacted]

Sincerely,

Katarzyna Ciazynska
(she/her)
Associate Editor
Nature Structural & Molecular Biology
<https://orcid.org/0000-0002-9899-2428>

Referee expertise:

Referee #1: transporters, biochemistry, pharmacology

Referee #2: transporters, pharmacology

Referee #3: transporters, cryo-EM

Reviewers' Comments:

Reviewer #1:

Remarks to the Author:

SGLT2 is a well-known glucose transporter as a target for newly developed diabetes drugs, SGLT2 inhibitors. Despite the importance of SGLT2, the transporter is notoriously difficult to express in cultured cells. In this study, the authors established a method to overexpress and purify the genuine hSGLT2-MAP17 complex and performed single-particle cryo-EM analyses of the complexes with five inhibitors. The authors obtained outward-facing structures with four gliflozin inhibitors. Importantly, the authors found that the complex with phlorizin, an original compound of all SGLT2 inhibitors on the market, showed an inward-open structure. The authors also visualized and characterized the Na⁺ binding site in the outward-facing structures. Their findings suggest a novel role for Na⁺ in the conformational change of the complex upon sugar uptake. This is also an important finding for understanding the mechanisms of many LeuT folded transporters. In addition, it would be beneficial to analyze their structures in detail with multiple inhibitors for further development of SGLT inhibitors, as well as inhibitors for some LeuT folded transporters.

The reviewer believes that this manuscript is worthy of publication in Nature Structure Biology. All the data presented are clear and convincing. However, there is a major concern about the structure of the manuscript, as described below.

Major concern

One of the most important findings in this manuscript is the characterization of the Na⁺ binding site. The authors discuss the role of Na⁺ in the binding site using the inward-opening structure even before showing the new inward-opening structure. This is very confusing. It would be better if the authors describe the inward-opening structure first and then show the new role of Na⁺.

Minor points

1. One is concerned that there are not many comparisons with previously reported structures by Niu et al 2022. Please consider visualizing the differences or similarities.

2. line 55; "Figs 3b" should be "Fig 3b".
3. line 68; "the LeuT transporter family" is rarely used. There are the LeuT fold transporters but no such a family.
4. line 123 and Fig. 1: "Fig. 1e" is not labeled in Fig.1.
5. Is it difficult to test substrates binding with and without Na⁺? Are there any previous reports?
6. lines 187-194; There are very interesting suggestions. However, the reviewer is afraid that it is too speculative without any data.
7. The authors discuss the selectivity of the inhibitors. There is no experimental result using mutants in the gliflozin binding site based on the author's findings. Any wet data provided by the authors or from previous studies can be added to the manuscript if possible.
8. line 234; Ref. 25 is probably a typo for this sentence.
9. lines 403-404 and Fig.1; Please clarify what a MAP-17 coding region and a hSGLT2-coding region. Are these the same plasmids as the plasmids used for purifications? If not the same, Fig 1d and (probably)1e may need to be moved from Fig 1. These data mislead readers as the authors tested their GFP contractions. There is no problem with using their constructs, but the authors can explain what they used.
10. line 746; "hSGLT2 without MAP17" is likely incorrect. The authors mentioned about hSGLT2-MAP17 heterodimer in the next line.
11. Figure 1a; It would be better to show both N- and C-terminals of MAP17 by dotted lines since the authors used the full length of MAP17.
12. Figure 1c; This panel is not systematic to indicate what the authors emphasize. If the authors want to show the merit of the "SP-sfGFP" strategy, it would be better to add "EGFP-SGLT2+MAP17" or "SGLT2-EGFP+MAP17". Or the authors can simply move the panel to the supplements together with d and "e".

Reviewer #2:

Remarks to the Author:

In the SGLT2/MAP17 complex, this paper described the binding structures of five gliflozin-type SGLT2 inhibitors and phlorizin by cryo-EM. It clarified the binding properties common to five gliflozin-type SGLT2 inhibitors. The authors identified the Na⁺ density in all compound-bound structures and discussed the conformational changes due to Na⁺ binding. The structure of SGLT2/MAP17 complex has already been solved, and the binding structures of SGLT2 inhibitors have already been published. Still, the results of this paper introduce a new perspective. The reviewers' comments are as follows:

General comments:

1. Since the Cryo-EM structure of inhibitor-bound SGLT2/MAP17 has already been

published, the abstract should clearly distinguish and state what is new in this paper and what this paper has confirmed in previous papers. Additionally, in lines 34-35, the content of "the unprecedented Na⁺-dependent sugar transport mechanism with MAP17 acting as a scaffold" needs to be explained briefly but more concretely in Abstract.

2.The authors describe that in addition to the Na₂ site, the Na₃ site of SGLT1 also stabilizes the outward-open, substrate-binding structure by connecting TM1 to TM5, as Na₂ connects TM1 and TM8 in SGLT2. It is not clearly explained how the site corresponding to Na₃ in SGLT2 is stabilized without Na binding. What interactions between residues are responsible for the stabilization without Na⁺-binding needs to be explained.

3.The reviewer understands that Na⁺-binding to the Na₂ site stabilizes the outward-facing, substrate-binding structure by connecting TM1 and TM8 in SGLT2. In the inward-open structure, because of the interaction between F98 and F453, Na₂ site is closed. For Na⁺ to bind to Na₂ site, the interaction between F98 and F453 should be release and Na₂ site has to open before Na⁺ binding. What force is energetically sufficient to move the inward open structure into the outward facing structure and release the interaction between F98 and F453? This needs to be clearly explained.

4.In the F98A and F453A mutants, it is interesting that they can transport alpha-MG, even though the interaction between F98 and F453, which contributes to stabilizing the inward-open structure, is disrupted. Does the instability of the inward-open structure in the F98A and F453A mutants not affect the transport cycle of alpha-MG transport? This needs to be explained.

Specific comments

5.Line 21: "Selective" is not necessary.

6.Line 45: "responsible for the reabsorption of plasma glucose in the proximal tubules" should be "responsible for the reabsorption of glucose in the proximal tubules"

7.Line 46: "renal capillary" should be "glomerulus".

8.Line 47: "absorbs 90% of plasma glucose" should be "absorbs 90% of filtered glucose"

9.Lines 171-172 "The binding affinity of canagliflozin to hSGLT2 was lower in the Na⁺ absent conditions (Fig. 2d, Supplementary Data Table 2)": Although the authors indicate The binding affinity of canagliflozin was lower in the Na⁺ absent condition, B_{max} decreased in the Na⁺ absent condition whereas K_d seems less affected (Supplementary Data Table 2). The authors emphasized the decrease of affinity and discussed structural features responsible for it. Thus, authors are requested to show statistical and biological significance of such a seemingly small alteration of K_d. Additionally, the authors should also intensively discuss the structural basis of the decrease of B_{max} in the Na⁺ absent condition.

10.In Fig. 2d, ordinate indicates the binding as % of B_{max}. It should be clearly indicated whether the values are normalized for the each B_{max} or the B_{max} of Na(+).

Reviewer #3:

Remarks to the Author:

SGLT2 reuptakes glucose in the kidney and is essential for glucose homeostasis. Therefore, SGLT2 is an important drug target in type II diabetes. The current work by Hiraizumi et al. determines the cryo-EM structures of human SGLT2 in complex with Canagliflozin, dapagliflozin, TA-1887, sotagliflozin, and phlorizin. The electron density maps are of high quality. The work is systematic, and comprehensive, and is a good complement to the previously published work on SGLT1 and SGLT2, including SGLT1 in the inward-open state (Han et al., *Nature*, 2021, PMID: 34880492), SGLT1+inhibitor LX2761 (Niu et al., *Nat Commun*, 2022, PMID: 36307403), and SGLT2+inhibitor Empagliflozin (Niu et al., *Nature*, 2021, PMID: 34880493).

I have a few major concerns and minor concerns which might be helpful for the authors to improve their manuscript.

Major issues:

1. It is surprising that the phlorizin-bound SGLT2 is in the inward-facing conformation. Functional data not only from other groups (Bisignano et al., *Nat Commun*, 2018, PMID: 30532032) but also presented in this manuscript suggest that phlorizin binds to the outward-open state with high-affinity. But why did the author not capture the phlorizin bound in the high affinity site but in the low-affinity site of inward-open SGLT2 in the current work?
2. Because of the high sequence and functional similarities between SGLT1 and SGLT2, it is hypothesized that they would share a similar transporting mechanism. The structures of SGLT1 in both outward-facing (Niu et al., *Nat Commun*, 2022, PMID: 36307403) and inward-facing states (Han et al., *Nature*, 2021, PMID: 34880492) are available already. The author may want to compare their inward-facing structure with that of SGLT1. Moreover, are the structural changes observed in SGLT2 here similar to that observed already in SGLT1 (Niu et al., *Nat Commun*, 2022, PMID: 36307403)?
3. In this work, the author observed that MAP17 does not change its conformation between states. But how does MAP17 enhance the activity of SGLT2? The author stated that MAP17 has the "scaffolding" function, which could not explain its mechanism. This point need to be elaborated with proper experimental data.

Minor issues:

1. The statement in the abstract need revision. Line 25: the mechanisms of SGLT2i are known from (Niu et al., *Nature*, 2021, PMID: 34880493) already.
2. Line 31-33: the functional role of sodium could be expected from previous work on vSGLT and other sodium-dependent transporters, such as LeuT, not "unexpected".
3. Fig. 2d: The K_d of canagliflozin (~ 100 nM) is much higher than its IC_{50} (~ 2.7 nM in literature) in the presence of sodium. In addition, sodium only increases the B_{max} but does not change the K_d too much. If both sodium and canagliflozin promote SGLT into the outward-facing state, sodium might decrease the K_d of canagliflozin. Additionally, the author might want to check if there was sodium contamination even in the sodium-free condition.
4. Fig. 2e,g: the difference between Sotagliflozin and Dapagliflozin is at the C5 position, leading to the loss of binding of S460 and T153 with sotagliflozin. The binding of Q457 to sotagliflozin is also decreased. These structural observations suggest that sotagliflozin should bind SGLT2 with much lower affinity compared with dapagliflozin. But why are there IC_{50} so close (Dominguez Rieg and Rieg, *Diabetes Obes. Metab.*, 2019, PMID: 31081587)? The authors might need to clarify these discrepancies.
5. In Fig. 3, the data suggests there is more than one site for phlorizin binding to WT SGLT2. However, if there are only two sites on SGLT2, the B_{max1} should be close to B_{max2} . But it seems like the low-affinity B_{max2} is much higher than B_{max1} in Fig.3d-f.

Are there additional phlorizin binding sites?

6. The sodium at the Na2 site was not observed for the phlorizin structure, is it possible due to its lower resolution (3.3Å), especially compared to other structures at higher resolution (2.8-3.1Å)? Alternatively, the author might want to reduce particle numbers in the dataset (eg. Canagliflozin dataset) to obtain a ~3.3 Å structure, and then check if the sodium could be observed in Na2 site in the 3.3 Å Canagliflozin map.

7. Positions of some waters need to be adjusted to better fit the map (eg. 4/HOH/D of Dapa complex).

8. Positions of some side chains need to be adjusted (eg. K567/A of Dapa ...).

9. There are obvious conformational changes of the intracellular loop (549-574 region) when Dapa complex is compared to the published SGLT2+Empa structure (Niu et al., Nature, 2021, PMID: 34880493) or SGLT1 structure (Han et al., Nature, 2021, PMID: 34880492). Authors might want to comment on this.

Author Rebuttal to Initial comments

Specific comments by Referee #1:

SGLT2 is a well-known glucose transporter as a target for newly developed diabetes drugs, SGLT2 inhibitors. Despite the importance of SGLT2, the transporter is notoriously difficult to express in cultured cells. In this study, the authors established a method to overexpress and purify the genuine hSGLT2-MAP17 complex and performed single-particle cryo-EM analyses of the complexes with five inhibitors. The authors obtained outward-facing structures with four gliflozin inhibitors. Importantly, the authors found that the complex with phlorizin, an original compound of all SGLT2 inhibitors on the market, showed an inward-open structure. The authors also visualized and characterized the Na⁺ binding site in the outward-facing structures. Their findings suggest a novel role for Na⁺ in the conformational change of the complex upon sugar uptake. This is also an important finding for understanding the mechanisms of many LeuT folded transporters. In addition, it would be beneficial to analyze their structures in detail with multiple inhibitors for further development of SGLT inhibitors, as well as inhibitors for some LeuT folded transporters.

The reviewer believes that this manuscript is worthy of publication in Nature Structure Biology. All the data presented are clear and convincing. However, there is a major concern about the structure of the manuscript, as described below.

Thank you for your positive comments. We are happy to hear your favorable evaluation.

Major concern

One of the most important findings in this manuscript is the characterization of the Na2 binding site. The authors discuss the role of Na⁺ in the binding site using the inward-opening structure even before showing the new inward-opening structure. This is very confusing. It would be better if the authors describe the inward-opening structure first and then show the new role of Na⁺.

Thank you for your pointing this out. We agree with your suggestion and have accordingly restructured the Results and Discussion as follows:

Original structure (subheadings):

1. Structural determination of the hSGLT2–MAP17 heterodimer
2. The inhibitor-bound hSGLT2 structure and the roles of the Na₂ site
3. Implication of the role of Na₃ site of SGLT1
4. The hSGLT2 outward-facing conformation in complex with inhibitors
5. High concentration phlorizin fixes hSGLT2 in the inward-opening state.
6. Structural rearrangement from the outward to inward conformations

Revised structure (subheadings)

1. Structural determination of the hSGLT2–MAP17 heterodimer
2. Comparison of hSGLT2 structures in outward-facing conformation with gliflozins
3. Subtype specificity between SGLT1 and SGLT2
4. High-concentration phlorizin fixes hSGLT2 in the inward-opening state
5. The role of Na⁺ in SGLT1/2
6. Structural rearrangement according to the alternating-access mechanism

Minor points

1. One is concerned that there are not many comparisons with previously reported structures by Niu et al 2022. Please consider visualizing the differences or similarities.

We have carefully compared our structures with those reported by Niu et al. (2022) and have added the following description (lines 164–174; Supplementary Fig. 10):

“The binding mode that we observed was”

2. line 55; “Figs 3b” should be “Fig 3b”.

Thank you for the correction. We have corrected this accordingly.

3. line 68; “the LeuT transporter family” is rarely used. There are the LeuT fold transporters but no such a family.

Thank you for the correction. We have revised this accordingly.

4. line 123 and Fig. 1: “Fig. 1e” is not labeled in Fig.1.

Thank you for the correction. We have corrected this accordingly.

5. *Is it difficult to test substrates binding with and without Na⁺? Are there any previous reports?*

It is difficult to compare sugar binding with and without Na⁺ because the Michaelis constant for glucose against SGLT2 is 5 mM (Hummel et al., Am. J. Physiol., Cell Physiol., 2011, PMID: 20980548), which is too weak for the purpose of analyzing direct binding. However, since SGLT1 is in dynamic equilibrium regardless of the presence or absence of sugar and Na⁺ (Paz et al., PNAS, 2018, PMID: 29507231), we believe that SGLT2 can adopt an outward-facing conformation in the absence of Na⁺ ions, enabling it to bind to sugars. For the same reason, canagliflozin binds even in the absence of Na⁺, to some extent (Fig. 2d). Nevertheless, sugar transport requires Na⁺ binding and an Na⁺ gradient across the cell membrane.

6. *lines 187-194; There are very interesting suggestions. However, the reviewer is afraid that it is too speculative without any data.*

Thank you for pointing this out. We have revised the description as follows (lines 328–335):

“The current outward-facing conformation of hSGLT2 exhibits”

7. *The authors discuss the selectivity of the inhibitors. There is no experimental result using mutants in the gliflozin binding site based on the author's findings. Any wet data provided by the authors or from previous studies can be added to the manuscript if possible.*

Thank you very much for pointing this out. Several experiments with mutants have been reported (Niu et al., Nature, 2022, PMID: 36307403; Niu et al., Nature, 2022, PMID: 34880493; Bisignano et al., Nat Commun, 2018, PMID: 30532032). We re-considered the structure around the sugar C6-OH of dapagliflozin and -SCH₃ of sotagliflozin, and explained why sotagliflozin acquired SGLT1 inhibitory activity. We also conducted 3D-RISM analysis of the S460 bound water molecules observed in this study, showing that they do not contribute to the binding affinity of the compounds. However, we have concluded that the subtype specificity of dapagliflozin cannot be well explained by structural comparison of the binding pocket; we have added a citation for the Asp268 mutation experiment (Bisignano, et al. Nat Commun, 2018) and have added the following text (lines 205-220):

“Dapagliflozin and sotagliflozin differ in the C₆ functional group,”

8. *line 234; Ref. 25 is probably a typo for this sentence.*

Thank you for pointing this out. We have corrected this as follows:

Nomura, S. et al. Novel indole-N-glucoside, TA-1887 as a sodium glucose cotransporter 2 inhibitor for treatment of type 2 diabetes. *ACS Med. Chem. Lett.* 5, 51–55 (2014)

9. *lines 403-404 and Fig.1; Please clarify what a MAP-17 coding region and a hSGLT2-coding region. Are these the same plasmids as the plasmids used for purifications? If not the same, Fig 1d and (probably) 1e may need to be moved from Fig 1. These data mislead readers as the authors tested their GFP contractions. There is no problem with using their constructs, but the authors can explain what they used.*

Thank you for pointing this out. We used the same plasmid for cryo-EM as for the binding and uptake assays. We have added a new section titled “**cDNA constructs**” to the Methods section.

10. line 746; “hSGLT2 without MAP17” is likely incorrect. The authors mentioned about hSGLT2–MAP17 heterodimer in the next line.

Thank you very much for pointing this out. We have corrected this accordingly.

11. Figure 1a; It would be better to show both N- and C-terminals of MAP17 by dotted lines since the authors used the full length of MAP17.

Thank you for the correction. We have corrected this accordingly.

12. Figure 1c; This panel is not systematic to indicate what the authors emphasize. If the authors want to show the merit of the “SP-sfGFP” strategy, it would be better to add “EGFP-SGLT2+MAP17” or “SGLT2-EGFP+MAP17”. Or the authors can simply move the panel to the supplements together with d and “e”.

We have added “EGFP-SGLT2+MAP17” and “SGLT2-EGFP+MAP17” in Figure 1C as shown below.

Figure 1c.

Specific comments by Referee #2:

In the SGLT2/MAP17 complex, this paper described the binding structures of five gliflozin-type SGLT2 inhibitors and phlorizin by cryo-EM. It clarified the binding properties common to five gliflozin-type SGLT2 inhibitors. The authors identified the Na⁺ density in all compound-bound structures and discussed the conformational changes due to Na⁺ binding. The structure of SGLT2/MAP17 complex has already been solved, and the binding structures of SGLT2 inhibitors have already been published. Still, the results of this paper introduce a new perspective. The reviewers' comments are as follows:

Thank you for your positive comments. We are happy to hear your favorable evaluation.

General comments:

1. Since the Cryo-EM structure of inhibitor-bound SGLT2/MAP17 has already been published, the abstract should clearly distinguish and state what is new in this paper and what this paper has confirmed in previous papers. Additionally, in lines 34-35, the content of "the unprecedented Na⁺-dependent sugar transport mechanism with MAP17 acting as a scaffold" needs to be explained briefly but more concretely in Abstract.

Thank you very much for pointing this out. We have accordingly revised the abstract as follows:

“Sodium-glucose cotransporter 2 (SGLT2) is important in glucose reabsorption. SGLT2 inhibitors suppress renal glucose reabsorption. Inhibiting SGLT2 therefore reduces blood glucose levels in type-2 diabetes patients. We and others have developed several SGLT2 inhibitors starting from phlorizin, a natural product. Using cryo-electron microscopy, we present the structures of human (h)SGLT2–MAP17 complexed with five natural or synthetic inhibitors. The four synthetic inhibitors (including canagliflozin) bind the transporter in the outward conformations, while phlorizin binds it in the inward conformation. The phlorizin–hSGLT2 interaction exhibits biphasic kinetics, suggesting that phlorizin alternately binds to the extracellular and intracellular sides. The Na⁺-bound outward-facing and unbound inward-opening structures of hSGLT2–MAP17 suggest that the MAP17-associated bundle domain functions as a scaffold, with the hash domain rotating around the Na⁺-binding site. Thus, Na⁺-binding stabilizes the outward-facing conformation and its release promotes state exchange to inward-open conformation, exhibiting a novel role of Na⁺ in symport mechanism. These results provide structural evidence for the Na⁺-coupled alternating-access mechanism proposed for the transporter family.”

Our current findings explain the biphasic inhibition of phlorizin and the proposed alternating-access model from a structural point of view.

2. The authors describe that in addition to the Na2 site, the Na3 site of SGLT1 also stabilizes the outward-open, substrate-binding structure by connecting TM1 to TM5, as Na2 connects TM1 and TM8 in SGLT2. It is not clearly explained how the site corresponding to Na3 in SGLT2 is stabilized without Na⁺ binding. What interactions between residues are responsible for the stabilization without Na⁺-binding needs to be explained.

Thank you for pointing this out. In the outward-facing conformation, the Na3 site is almost identical between SGLT1 and SGLT2. The residue T395 in SGLT1, which constitutes the Na3 site, is changed to A395 in SGLT2. However, in our current study, we were unable to identify any residues responsible for stabilizing the Na3 site in SGLT2 in the absence of Na⁺. We have revised the description as follows (lines

319–335) and added figure4a-d:

Figure 4a-d.

“SGLT1 requires two Na⁺ ions for sugar transport ...”

3. The reviewer understands that Na⁺-binding to the Na2 site stabilizes the outward-facing, substrate-binding structure by connecting TM1 and TM8 in SGLT2. In the inward-open structure, because of the interaction between F98 and F453, Na2 site is closed. For Na⁺ to bind to Na2 site, the interaction between F98 and F453 should be released and Na2 site has to open before Na⁺ binding. What force is energetically sufficient to move the inward open structure into the outward facing structure and release the interaction between F98 and F453? This needs to be clearly explained.

Thank you for pointing this out. Energy landscape studies using MD simulation have shown that inward-open is the most stable conformation in the absence of Na⁺ and substrate, whereas from the perspective of membrane potential, outward-facing Na⁺ binding is favored (Paz et al., PNAS, 2018, PMID: 29507231). We therefore consider membrane potential to be the driving force behind the release of the F98–F453 interaction. We have accordingly revised the description as follows (lines 388-398):

“Although this study does not fully elucidate the dynamics of Na⁺ binding/release ...”

Additionally, the structural transition from inward-open to outward-facing has been investigated in detail in the LeuT transporters (Malinauskaite, Nat Comm. 2016, PMID: 27221344), revealing that the Leu25 side-chain in the path of the substrate acts as a gate that opens to the outward conformation by filling the void after Na⁺ is released. In SGLT2, Leu25 of LeuT corresponds to Ile76, which is in fact conserved. However, compared with SGLT1, the outward- and inward-facing structures of SGLT2 do not exhibit main-

chain flipping involving Leu25. Instead, TM8 moves throughout to create a path. We therefore think that there is a different structural mechanism for the switch from inward- to outward-facing conformation.

4. In the F98A and F453A mutants, it is interesting that they can transport alpha-MG, even though the interaction between F98 and F453, which contributes to stabilizing the inward-open structure, is disrupted. Does the instability of the inward-open structure in the F98A and F453A mutants not affect the transport cycle of alpha-MG transport? This needs to be explained.

Thank you for pointing this out. The F98A and F453A mutations impair α -MG transport activity. (Fig. 3h). Substrate transport among the LeuT transporters such as SGLT2 has been described as “alternating access” (Forrest et al. Physiology 2009, PMID: 19996368), according to which substrates are taken up in the outward conformation, followed by complete closure of the outer gate and uptake into the cytosol via the inward conformation. For this alternating access to function correctly, it is essential that there is no leakage of Na^+ or substrate from outside the cell when in the inward conformation. The interaction between F98 and F453 is part of a wall that blocks access from the outside of the cell when the inward conformation is adopted. By changing F98 or F453 to Ala, this wall becomes much thinner. We believe that these mutations destabilize the inward structure, thereby reducing alternating-access transport activity.

Specific comments

5. Line 21: “Selective” is not necessary.

Thank you for the correction. We have corrected this accordingly.

6. Line 45: “responsible for the reabsorption of plasma glucose in the proximal tubules” should be “responsible for the reabsorption of glucose in the proximal tubules”

Thank you for the correction. We have corrected this accordingly.

7. Line 46: “renal capillary” should be “glomerulus”.

Thank you for the correction. We have corrected this accordingly.

8. Line 47: “absorbs 90% of plasma glucose” should be “absorbs 90% of filtered glucose”

Thank you for the correction. We have corrected this accordingly.

9. Lines 171-172 “The binding affinity of canagliflozin to hSGLT2 was lower in the Na^+ absent conditions (Fig. 2d, Supplementary Data Table 2)”: Although the authors indicate The binding affinity of canagliflozin was lower in the Na^+ absent condition, B_{max} decreased in the Na^+ absent condition whereas K_d seems less affected (Supplementary Data Table 2). The authors emphasized the decrease of affinity and discussed structural features responsible for it. Thus, authors are requested to show statistical and biological significance of such a seemingly small alteration of K_d . Additionally, the authors should also intensively discuss the structural basis of the decrease of B_{max} in the Na^+ absent condition.

Thank you for pointing this out. SGLT2 is reported to be in dynamic equilibrium in the presence or absence of substrate and Na^+ (Paz et al., PNAS, 2018, PMID: 29507231). Thus, even in the absence of Na^+ , a certain proportion of the transporter adopts the outward-conformation; Na^+ increases this proportion, thereby

altering the B_{max} of canagliflozin. We added Na^+ concentration-dependent change in the amount of canagliflozin bound (Supplementary Fig. 15a), showing that Na^+ -dependent shift to outward-conformation occurs. Additionally, the B_{max} of TA-1887 toward hSGLT2 was also lower in the absence than in the presence of Na^+ (Supplementary Fig. 15b).

Supplementary Fig. 15

We believe that these results explain why Na^+ binding has not always been observed in other SGLTs in the reported structures (Niu et al., Nat Commun, 2022, PMID: 36307403, Niu et al., Nature, 2021, PMID: 34880493). We have revised “the affinity” to “the maximum number of binding sites” and have revised the text. Although some report showed that the affinity for substrates and inhibitor is decreased in LeuT fold transporters by the mutation of Na2 site (Bisignano et al., Nat Commun, 2022, PMID 30532032, Khafizov et al., PNAS, 2012, PMID 23047697), our study showed K_d did not vary in the presence or absence of Na^+ . It is possible that the use of membrane fractions with no Na^+ concentration gradient and potential gradient resulted in only stabilization to outward-conformation as reported (Paz et al., PNAS, 2018, PMID: 29507231, Wahlgren et al., Nat Commun, 2018, PMID 29717135), without affecting K_d .

10. In Fig. 2d, ordinate indicates the binding as % of B_{max} . It should be clearly indicated whether the values are normalized for the each B_{max} or the B_{max} of $\text{Na}(+)$.

Thank you for the correction. We have revised the text accordingly.

Specific comments by Referee #3:

SGLT2 reuptakes glucose in the kidney and is essential for glucose homeostasis. Therefore, SGLT2 is an important drug target in type II diabetes. The current work by Hiraizumi et al. determines the cryo-EM structures of human SGLT2 in complex with Canagliflozin, dapagliflozin, TA-1887, sotagliflozin, and phlorizin. The electron density maps are of high quality. The work is systematic, and comprehensive, and is a good complement to the previously published work on SGLT1 and SGLT2, including SGLT1 in the inward-open state (Han et al., Nature, 2021, PMID: 34880492), SGLT1+inhibitor LX2761 (Niu et al., Nat Commun, 2022, PMID: 36307403), and SGLT2+inhibitor Empagliflozin (Niu et al., Nature, 2021, PMID: 34880493). I have a few major concerns and minor concerns which might be helpful for the authors to improve their manuscript.

Thank you for your positive comments. We are happy to hear your favorable evaluation.

Major issues:

1. It is surprising that the phlorizin-bound SGLT2 is in the inward-facing conformation. Functional data not only from other groups (Bisignano et al., Nat Commun, 2018, PMID: 30532032) but also presented in this manuscript suggest that phlorizin binds to the outward-open state with high-affinity. But why did the author not capture the phlorizin bound in the high affinity site but in the low-affinity site of inward-open SGLT2 in the current work?

Thank you for pointing this out. In the absence of a Na⁺ gradient in the cell extract, SGLT has been demonstrated to be in dynamic equilibrium between the inward and outward states, regardless of the presence or absence of substrate (Paz et al., PNAS, 2018, PMID: 29507231). In Fig. 3d, we show that at a high concentration of phlorizin, inward binding is more frequent than outward binding, suggesting that inward binding is favored under these conditions. We have corrected the text accordingly (lines 255-267):

“Phlorizin at up to 5000 nM did not saturate t....”

2. Because of the high sequence and functional similarities between SGLT1 and SGLT2, it is hypothesized that they would share a similar transporting mechanism. The structures of SGLT1 in both outward-facing (Niu et al., Nat Commun, 2022, PMID: 36307403) and inward-facing states (Han et al., Nature, 2021, PMID: 34880492) are available already. The author may want to compare their inward-facing structure with that of SGLT1. Moreover, are the structural changes observed in SGLT2 here similar to that observed already in SGLT1 (Niu et al., Nat Commun, 2022, PMID: 36307403)?

Thank you for pointing this out. The SGLT1 inward-open conformation, lacking ligand binding, exhibits less inward movement than that of SGLT2. Nonetheless, the outward and inward changes discussed here occur at the same TM and are thought to undergo similar conformational changes. We have added Fig5a,b and a discussion of conformational change between the inward-open and outward-facing structures (lines 338–347):

Figure 5a, b

“In the inward-facing conformation, reported for vSGLT and hSGLT1”

3. In this work, the author observed that MAP17 does not change its conformation between states. But how does MAP17 enhance the activity of SGLT2? The author stated that MAP17 has the “scaffolding” function, which could not explain its mechanism. This point need to be elaborated with proper experimental data.

Thank you for pointing this out. On further consideration, we determined that our original statement, “MAP17 is a scaffold,” was an overstatement and have revised it as follows:

“The Na⁺-bound outward-facing and unbound inward-opening structures of hSGLT2–MAP17 suggest that the MAP17-associated bundle domain functions as a scaffold, with the hash domain rotating around the Na⁺-binding site” (in the Abstract).

For LeuT transporters, a rocking-bundle model has been proposed, in which there are two domains for substrate transport, with the rocking bundle alternately opening inwards and outwards (Forrest et al., PNAS, 2009, PMID: 19996368). We consider that MAP17 forms part of the immobile scaffold. An additional experiment revealed that the melting temperature of SGLT2 was higher under MAP17 co-expression than in the absence of MAP17 (below; Supplementary Fig. 1d), indicating that MAP17 has a stabilizing effect on SGLT2.

Supplementary Fig. 1d

We have therefore added the following text (lines 351–362).

“In addition to In addition to SGLT2’s TMs,....”

Minor issues:

1. The statement in the abstract need revision. Line 25: the mechanisms of SGLT2i are known from (Niu et al., Nature, 2021, PMID: 34880493) already.
2. Line 31-33: the functional role of sodium could be expected from previous work on vSGLT and other sodium-dependent transporters, such as LeuT, not “unexpected”.

Thank you for pointing this out. We have revised the Abstract as follows:

“Sodium-glucose cotransporter 2 (SGLT2) is important in glucose reabsorption. SGLT2 inhibitors suppress renal glucose reabsorption. Inhibiting SGLT2 therefore reduces blood glucose levels in type-2 diabetes patients. We and others have developed several SGLT2 inhibitors starting from phlorizin, a natural product. Using cryo-electron microscopy, we present the structures of human (h)SGLT2–MAP17 complexed with five natural or synthetic inhibitors. The four synthetic inhibitors (including canagliflozin) bind the transporter in the outward conformations, while phlorizin binds it in the inward conformation. The phlorizin–hSGLT2 interaction exhibits biphasic kinetics, suggesting that phlorizin alternately binds to the extracellular and intracellular sides. The Na⁺-bound outward-facing and unbound inward-opening structures of hSGLT2–MAP17 suggest that the MAP17-associated bundle domain functions as a scaffold, with the hash domain rotating around the Na⁺-binding site. Thus, Na⁺-binding stabilizes the outward-facing conformation and its release promotes state transition to inward-open conformation, exhibiting a novel role of Na⁺ in symport mechanism. These results provide structural evidence for the Na⁺-coupled alternating-access mechanism proposed for the transporter family.”

3. Fig. 2d: The K_d of canagliflozin (~ 100 nM) is much higher than its IC_{50} (~ 2.7 nM in literature) in the presence of sodium. In addition, sodium only increases the B_{max} but does not change the K_d too much. If both sodium and canagliflozin promote SGLT into the outward-facing state, sodium might decrease the K_d of canagliflozin. Additionally, the author might want to check if there was sodium contamination even in the sodium-free condition.

Thank you for pointing this out. We consider that the differences between the IC_{50} and K_d of canagliflozin are due to differences in the experimental conditions. That is, the glucose uptake assay was conducted under Na^+ gradients, whereas the binding assay was conducted in the absence of membrane potential and Na^+ gradients. Similar differences have been reported for empagliflozin (Grenpler, et al. 2012, PMID: 21985634). Furthermore, SGLT2 is reported to be in dynamic equilibrium regardless of the presence or absence of the substrate or Na^+ (Paz et al., PNAS, 2018, PMID: 29507231). Thus, even in the absence of Na^+ , a certain proportion of the transporter adopts the outward conformation. We added Na^+ concentration-dependent change in the amount of canagliflozin bound (Supplementary Fig. 15a), showing that Na^+ -dependent shift to outward-conformation occurs. Additionally, the B_{max} of TA-1887 toward hSGLT2 was also lower in the absence than in the presence of Na^+ (Supplementary Fig. 15b).

Supplementary Fig. 15

We believe that these results explain why Na^+ binding has not always been observed in other SGLTs in the reported structures (Niu et al., Nat Commun, 2022, PMID: 36307403, Niu et al., Nature, 2021, PMID: 34880493). Although some report showed that the affinity for substrates and inhibitor is decreased in LeuT fold transporters by the mutation of Na2 site (Bisignano et al., Nat Commun, 2022, PMID 30532032, Khafizov et al., PNAS, 2012, PMID 23047697), our study showed K_d did not vary in the presence or absence of Na^+ . It is possible that the use of membrane fractions with no Na^+ concentration gradient and potential gradient resulted in only stabilization to outward-conformation as reported (Paz et al., PNAS, 2018, PMID: 29507231, Wahlgren et al., Nat Commun, 2018, PMID 29717135), without affecting K_d .

4. Fig. 2e,g: the difference between Sotagliflozin and Dapagliflozin is at the C5 position, leading to the loss of binding of S460 and T153 with sotagliflozin. The binding of Q457 to sotagliflozin is also decreased. These structural observations suggest that sotagliflozin should bind SGLT2 with much lower affinity compared with dapagliflozin. But why are their IC50 so close (Dominguez Rieg and Rieg, *Diabetes Obes. Metab.*, 2019, PMID: 31081587)? The authors might need to clarify these discrepancies.

Thank you very much for pointing this out. Several experiments with mutants have been reported (Niu et al., *Nature*, 2022, PMID: 36307403; Niu et al., *Nature*, 2022, PMID: 34880493; Bisignano et al., *Nat Commun*, 2018, PMID: 30532032). We re-considered the structure around the sugar C6-OH of dapagliflozin and -SCH₃ of sotagliflozin, and explained why sotagliflozin acquired SGLT1 inhibitory activity. We also conducted 3D-RISM analysis of the S460 bound water molecules observed in this study, showing that they do not contribute to the binding affinity of the compounds. However, we have concluded that the subtype specificity of dapagliflozin cannot be well explained by structural comparison of the binding pocket; we have added a citation for the Asp268 mutation experiment (Bisignano, et al. *Nat Commun*, 2018) and have added the following text (lines 205-220):

“Dapagliflozin and sotagliflozin differ in the C₆ functional group”

5. In Fig. 3, the data suggests there is more than one site for phlorizin binding to WT SGLT2. However, if there are only two sites on SGLT2, the B_{max1} should be close to B_{max2} . But it seems like the low-affinity B_{max2} is much higher than B_{max1} in Fig.3d-f. Are there additional phlorizin binding sites?

Thank you very much for pointing this out. The large difference between B_{max1} and B_{max2} exists because binding to the intracellular site (reflected by B_{max2}) is favored at the high concentration of phlorizin. Further, given that there are two slopes in Fig. 3f, and based on the evidence provided by our structural analysis, the binding site is considered to comprise two sites.

6. The sodium at the Na2 site was not observed for the phlorizin structure, is it possible due to its lower resolution (3.3Å), especially compared to other structures at higher resolution (2.8-3.1Å)? Alternatively, the author might want to reduce particle numbers in the dataset (eg. Canagliflozin dataset) to obtain a ~3.3 Å structure, and then check if the sodium could be observed in Na2 site in the 3.3 Å Canagliflozin map.

Thank you very much for pointing this out. In the phlorizin-bound inward-open structure, phlorizin is bound to the Na2 site, TM1 and TM8 are separate, and Na⁺ is not within the coordination distance (Supplementary Fig. 11c, d).

7. Positions of some waters need to be adjusted to better fit the map (eg. 4/HOH/D of Dapa complex).
8. Positions of some side chains need to be adjusted (eg. K567/A of Dapa ...).

Thank you very much for pointing this out. We have revised the model, re-registered it in the PDB, and attached a validation report.

9. There are obvious conformational changes of the intracellular loop (549-574 region) when Dapa complex is compared to the published SGLT2+Empa structure (Niu et al., *Nature*, 2021, PMID: 34880493) or SGLT1 structure (Han et al., *Nature*, 2021, PMID: 34880492). Authors might want to comment on this.

Thank you very much for pointing this out. We have carefully compared our structures with those reported by Niu et al. (2022) and now describe this in the revised text as follows (lines 166–174); we have also added a panel to Supplementary Fig. 10.

Supplementary Fig. 10

“When the overall structures (including those of MAP17).....”

Decision Letter, first revision:

Message: Our ref: NSMB-A47244A

15th Jun 2023

Dear Dr. Miyaguchi,

Thank you for submitting your revised manuscript "Structural insights into the mechanism of the human SGLT2-MAP17 glucose transporter" (NSMB-A47244A). It has now been seen by the original referees and their comments are below. The reviewers find that the paper has improved in revision, and therefore we'll be happy in principle to publish it in Nature Structural & Molecular Biology, pending minor revisions to satisfy the referees' final requests and to comply with our editorial and formatting guidelines.

Sincerely,

Katarzyna Ciazynska
(she/her)
Associate Editor
Nature Structural & Molecular Biology
<https://orcid.org/0000-0002-9899-2428>

Reviewer #1 (Remarks to the Author):

All the issues raised in my initial review have been addressed by the authors in a commendable manner. They have made important changes to the structure of the manuscript, thereby improving the narrative flow. Their restructured manuscript, coupled with their new findings on the SGLT2-MAP17 complex and the role of Na⁺, provides a clearer and more concise picture of their findings.

The authors have also effectively addressed the minor points, such as the need for comparison with previous research. Their detailed comparison with the structures reported by Niu et al. (2022) adds considerable depth to their study. Furthermore, the corrections of typographical errors and figure captions have improved the readability of the manuscript. In addition, the authors clarified experimental challenges and provided detailed explanations regarding substrate binding with and without Na⁺. They have also addressed concerns about speculation by refining the narrative, thus providing a more informed interpretation of their results.

Overall, the authors thoroughly and satisfactorily addressed all concerns raised in the initial review. Based on the revisions and the importance of the study, I support the publication of this manuscript in Nature Structural and Molecular Biology.

Reviewer #2 (Remarks to the Author):

No additional comments to authors.

Reviewer #3 (Remarks to the Author):

This revision answered most of my questions and substantially strengthened the manuscript. I have some remaining questions:

1. The authors cited (Paz et al., PNAS, 2018, PMID: 29507231) in their rebuttal letter to provide evidence of the presence of both inward-facing and outward-facing state conformations in the absence of substrate. I am wondering why phlorizin prefers to stabilize the protein in the inward-facing state while the other inhibitors prefer the outward-facing state. Moreover, there are many pieces of evidence showing that phlorizin mainly inhibits SGLTs in the outward-facing state (Wahlgren et al., Nat Commun, 2018, PMID: 29717135).
2. The authors have provided evidence showing MAP17 could enhance the thermostability

of SGLT2 in the revision. However, thermostability is not always correlated with transporter activity. How MAP17 activates SGLT2 seems still unknown. The author might comment on this.

3. Line 110: MAP17 is not a scaffold as indicated in the rebuttal letter. The author might want to rephrase this sentence.

Ref:

Wahlgren, W.Y., Dunevall, E., North, R.A., Paz, A., Scalise, M., Bisignano, P., Bengtsson-Palme, J., Goyal, P., Claesson, E., Caing-Carlsson, R., et al. (2018). Substrate-bound outward-open structure of a Na(+)-coupled sialic acid symporter reveals a new Na(+) site. Nat Commun 9, 1753.

Author Rebuttal, first revision:

Specific comments by Referee #1:

All the issues raised in my initial review have been addressed by the authors in a commendable manner. They have made important changes to the structure of the manuscript, thereby improving the narrative flow. Their restructured manuscript, coupled with their new findings on the SGLT2-MAP17 complex and the role of Na⁺, provides a clearer and more concise picture of their findings.

The authors have also effectively addressed the minor points, such as the need for comparison with previous research. Their detailed comparison with the structures reported by Niu et al. (2022) adds considerable depth to their study. Furthermore, the corrections of typographical errors and figure captions have improved the readability of the manuscript. In addition, the authors clarified experimental challenges and provided detailed explanations regarding substrate binding with and without Na⁺. They have also addressed concerns about speculation by refining the narrative, thus providing a more informed interpretation of their results.

Overall, the authors thoroughly and satisfactorily addressed all concerns raised in the initial review. Based on the revisions and the importance of the study, I support the publication of this manuscript in Nature Structural and Molecular Biology.

We appreciate your evaluation.

Specific comments by Referee #3:

1. The authors cited (Paz et al., PNAS, 2018, PMID: 29507231) in their rebuttal letter to provide evidence of the presence of both inward-facing and outward-facing state conformations in the absence of substrate. I am wondering why phlorizin prefers to stabilize the protein in the inward-facing state while the other inhibitors prefer the outward-facing state. Moreover, there are many pieces of evidence showing that phlorizin mainly inhibits SGLTs in the outward-facing state (Wahlgren et al., Nat Commun, 2018, PMID: 29717135).

As explained in lines: 260–264, it has been reported that phlorizin inhibits from within the cell when there is no Na⁺ concentration gradient and at high phlorizin concentrations. In the present

study, the kinetics analysis using membrane fraction showing Figs. 3d–f indicated that binding from the inside predominates for the reasons above. Moreover, unlike gliflozin, phlorizin contains an ethyl bond, which allows it to bind from the inside (lines: 253–255).

2. *The authors have provided evidence showing MAP17 could enhance the thermostability of SGLT2 in the revision. However, thermostability is not always correlated with transporter activity. How MAP17 activates SGLT2 seems still unknown. The author might comment on this.*

As SGLT2 can be expressed in the plasma membrane without MAP17 but requires MAP17 to function, MAP17 was thought to contribute to the active conformation of SGLT2 (J. Am. Soc. Nephrol, 2017, PMID: 27288013). However, as SGLT2 is almost identical to SGLT1 (which does not require MAP17 for activity) structurally and there is no evidence that MAP17 is directly involved in sugar transport activity, we investigated its thermostability in the membrane. In addition, it was recently reported that MAP17 is involved in the localization of SGLT2 to membrane surface (Nat comm, 2023, PMID: 37217492).

We consider that co-expression of SGLT2 with MAP17 optimizes its localization and/or folding at the plasma membrane for stability, leading to its functional expression.

We modified the manuscript to include these points (lines: 359–369).

3. *Line 110: MAP17 is not a scaffold as indicated in the rebuttal letter. The author might want to rephrase this sentence.*

We apologize for the oversight. Based on the structural and experimental results, we believe that MAP17 is part of the scaffolding of the rocking bundle, together with the bundle domain, as described in lines 361–362. We removed the term “a scaffold” in line 111.

Final Decision Letter:

Message 22nd Sep 2023

:

Dear Dr. Miyaguchi,

We are now happy to accept your revised paper "Structural insights into the mechanism of the human SGLT2–MAP17 glucose transporter" for publication as an Article in Nature Structural & Molecular Biology.

Acceptance is conditional on the manuscript's not being published elsewhere and on there

being no announcement of this work to the newspapers, magazines, radio or television until the publication date in Nature Structural & Molecular Biology.

Your paper will be published online soon after we receive proof corrections and will appear in print in the next available issue. You can find out your date of online publication by contacting the production team shortly after sending your proof corrections. Content is published online weekly on Mondays and Thursdays, and the embargo is set at 16:00 London time (GMT)/11:00 am US Eastern time (EST) on the day of publication. Now is the time to inform your Public Relations or Press Office about your paper, as they might be interested in promoting its publication. This will allow them time to prepare an accurate and satisfactory press release. Include your manuscript tracking number (NSMB-A47244B) and our journal name, which they will need when they contact our press office.

About one week before your paper is published online, we shall be distributing a press release to news organizations worldwide, which may very well include details of your work. We are happy for your institution or funding agency to prepare its own press release, but it must mention the embargo date and Nature Structural & Molecular Biology. If you or your Press Office have any enquiries in the meantime, please contact press@nature.com.

Please note that *Nature Structural & Molecular Biology* is a Transformative Journal (TJ). Authors may publish their research with us through the traditional subscription access route or make their paper immediately open access through payment of an article-processing charge (APC). Authors will not be required to make a final decision about access to their article until it has been accepted. [Find out more about Transformative Journals](https://www.springernature.com/gp/open-research/transformative-journals)

Authors may need to take specific actions to achieve [compliance with funder and institutional open access mandates](https://www.springernature.com/gp/open-research/funding/policy-compliance-faqs). If your research is supported by a funder that requires immediate open access (e.g. according to [Plan S principles](https://www.springernature.com/gp/open-research/plan-s-compliance)) then you should select the gold OA route, and we will direct you to the compliant route where possible. For authors selecting the subscription publication route, the journal's standard licensing terms will need to be accepted, including [self-archiving policies](https://www.springernature.com/gp/open-research/policies/journal-policies). Those licensing terms will supersede any other terms that the author or any third party may assert apply to any version of the manuscript.

Sincerely,

Katarzyna Ciazynska
(she/her)
Associate Editor
Nature Structural & Molecular Biology
<https://orcid.org/0000-0002-9899-2428>

Click here if you would like to recommend Nature Structural & Molecular Biology to your librarian:

<http://www.nature.com/subscriptions/recommend.html#forms>